# Confidence as Control: A Survey of Confidence Utilization in Large Language Models

## Abstract

Most work on confidence in large language models has focused on estimation, uncertainty quantification, and calibration. In deployed systems, however, the key question is how confidence should be used to govern behavior. This survey studies **confidence utilization**: the use of confidence-related signals to control system decisions. We formalize this perspective through a unified framework in which confidence is defined over decision units under a local state and then consumed by a policy to determine actions. Using this lens, we organize the literature across the full LLM lifecycle: training, inference, model selection and cascading, retrieval-augmented generation, risk management, and agentic control. We compare methods by signal source, decision unit, and functional role, and conclude by highlighting open challenges in confidence semantics, composition, source attribution, decision-aware evaluation, and robustness. Overall, the survey positions confidence not only as an estimation target, but as a control primitive: a signal that systems already use to reduce cost at matched accuracy, to enforce coverage guarantees, and to trade answer rate against error rate—and whose principled use we argue is a prerequisite for reliable LLM systems.

## 1 Introduction

Confidence in Large Language Models (LLMs) bridges the gap between internal uncertainty and actionable system behavior. A model that knows what it doesn't know is useful; a system that acts on this knowledge—retrieving when uncertain, deferring when unreliable, focusing learning where needed—is transformative. This survey concerns the latter. In this survey, we use *confidence* broadly to denote signals about the expected reliability or usefulness of a model decision, including uncertainty estimates, log-probability-based scores, verbalized confidence, sample agreement, semantic uncertainty, and verifier or reward-model scores. Recent work has shown that LLMs can expose useful self-knowledge through formulations such as $P(\text{True})$ and $P(\text{IK})$ (Kadavath et al., 2022); that RLHF-tuned models can often express better-calibrated verbalized confidence when prompted appropriately (Tian et al., 2023); that semantic entropy over meaning clusters helps detect confabulations in free-form generation (Farquhar et al., 2024); and that agreement across sampled outputs can serve as a practical reliability proxy in reasoning and black-box factuality checking (Wang et al., 2023a; Manakul et al., 2023). These advances have made confidence estimation increasingly usable in practice.

At the same time, several surveys and benchmarking studies have examined confidence and uncertainty in LLMs, predominantly focusing on estimation and calibration. Geng et al. (2024) provide a foundational taxonomy of confidence estimation and calibration techniques, while Shorinwa et al. (2025) and Liu et al. (2025c) offer comprehensive coverage of uncertainty quantification methods and their applications. Xie et al. (2024) focus specifically on the calibration process for black-box LLMs, and Xiong et al. (2024) provide a systematic empirical evaluation of black-box confidence elicitation strategies. Taken together, this literature largely asks: how can we estimate, elicit, or calibrate high-quality confidence signals? Yet a good confidence score is not the end goal; it is the prerequisite. Recent surveys have begun to identify the shift from estimation to use: Zhang et al. (2026) chart the evolution of uncertainty quantification from a passive diagnostic into an active signal, organized around three application frontiers—reasoning, autonomous agents, and reinforcement learning. Our survey shares that motivation but takes an operational systems view: we formalize confidence utilization as a policy over decision units and follow it across the full LLM lifecycle—including

routing and cascading and retrieval-augmented generation, which are absent from their taxonomy, and risk management, which we develop as a standalone operational domain with deployment-grade guarantees (Appendix A details the relationship). A lifecycle-wide operational treatment of how confidence signals are consumed by downstream policies has, to our knowledge, been missing; this survey aims to provide one.

Our survey addresses the critical next step: **how should systems utilize confidence?** As illustrated in Figure 1, we propose a taxonomy where confidence functions as *control*—not a passive measure of uncertainty, but an active governor that determines what to learn, how to reason, and when to defer (Figure 4 maps the surveyed methods onto this taxonomy at the method level). We trace this utilization across the full LLM lifecycle: (i) training-time applications in data curation and alignment (§3), (ii) inference-time control over reasoning and decoding (§4), (iii) deployment-time decisions including model selection (§5), retrieval-augmented generation (RAG) (§6), and risk management (§7), and (iv) *agentic* settings (§8), where confidence becomes an actual control signal in multi-step loops: gating tool use and retries, triggering iterative refinement and backtracking, pruning search trees via verifier/process-reward confidence, and weighting votes in multi-agent debate. By synthesizing these distinct domains, we aim to offer a unified perspective for designing systems that act on uncertainty rather than merely quantify it. Two features distinguish this survey from a catalog. First, §2 states explicit inclusion criteria (C1–C3) that determine what counts as confidence utilization, distinguishes core from adjacent work in every comparison table, and documents how the corpus was assembled. Second, we use the resulting source/unit/role decomposition analytically: §2.3 derives structural constraints, cross-domain equivalences, and exposed contradictions from it, and Figure 3 populates the design space to identify combinations that are underrepresented in our corpus—candidate research gaps.

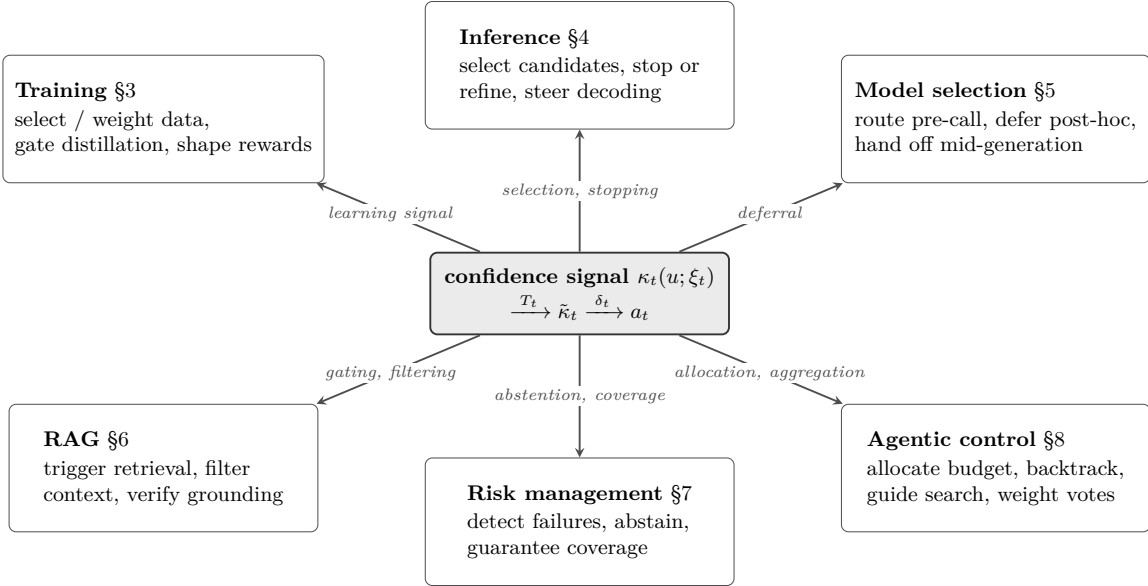

Figure 1: Confidence utilization across six domains of the LLM lifecycle. A confidence signal $\kappa_t(u; \xi_t)$, optionally transformed by $T_t$, is consumed by a policy $\delta_t$ whose action differs by stage: it allocates gradient mass at training time, selects and stops at inference time, defers across models, gates retrieval, controls abstention and coverage, and steers search and aggregation in agents. Section numbers indicate where each family is treated.

## 2 Unified Definition, Scope, and Notation

To compare methods across training, inference, routing, retrieval, risk control, and agentic systems, we adopt a single abstraction in which confidence is treated as a control signal. Throughout this survey, *confidence* is used in a deliberately broad operational sense: it may be a probability-like estimate, an uncertainty score,

a log-probability-derived quantity, a verbalized confidence value, a sample-agreement statistic, a semantic uncertainty measure, a verifier or judge score, a reward-model score, or a hybrid of several such signals. This breadth is a scoping decision, not a semantic claim: we do *not* assert that these signals estimate the same underlying quantity, and §2.2 gives the explicit test that determines which of them fall inside the survey and which are treated as adjacent. What the abstraction unifies is the *control interface*—how a reliability-relevant score enters a decision—rather than the semantics of the score itself, in the same way that a Markov decision process unifies problems whose rewards measure incommensurable quantities.

**Terminology.** We fix three terms and use them consistently. *Confidence utilization* names the object of study: the consumption of confidence-related signals by downstream decisions. *Confidence-as-control* names the framework of this section: the abstraction that expresses any utilization method as a signal–transformation–policy–action pipeline. When we describe an individual mechanism as *confidence-guided* (e.g., confidence-guided model selection), the adjective refers to a specific instantiation of that pipeline in one domain. The three terms therefore sit at three levels of abstraction—field of study, framework, and mechanism—and are not interchangeable.

## 2.1 Notation

Formally, let $\xi_t$ denote the *decision state* at stage $t$. This state represents the full local context in which the system acts, and may include the input query, a partial generation, sampled candidates or reasoning traces, retrieved evidence, available tools or models, environment observations, memory, or a remaining compute budget. Let $U_t = \{u_1, \ldots, u_n\}$ denote the set of *decision units* currently under consideration. A unit $u$ may be a token, span, claim, retrieved chunk, training example, candidate response, model, tool, reasoning step, trajectory, or agent vote. This notation is intentionally more general than the conventional pair $(x, y)$ because many confidence-utilization methods do not operate solely on completed outputs.

A *confidence signal* is a score

$$\kappa_t(u; \xi_t) \in \mathbb{R}^m$$

assigned to a unit $u$ under state $\xi_t$. Most methods use a scalar signal, but allowing $\kappa_t$ to be vector-valued is convenient for systems that combine several sub-signals before acting. Crucially, every signal in this survey is an estimator of some *target property* of the unit. For core methods the target property is binary, $\rho(u) \in \{0, 1\}$—correctness of an answer, groundedness of a claim, future success of a partial trajectory, adequacy of a model for a query—and $\kappa_t$ is intended to be monotonically related to $\Pr[\rho(u) = 1 \mid \xi_t]$. Adjacent methods (§2.2) instead estimate a real-valued property $\nu(u)$—expected utility, difficulty, data influence, or generic quality—that is correlated with, but not identical to, such a reliability property. Different methods choose different target properties, and much of the apparent heterogeneity of the literature dissolves once the target property is stated explicitly.

Confidence *utilization* is then the induced decision process

$$a_t \sim \delta_t(\cdot \mid \xi_t, U_t, \kappa_t), \qquad \xi_{t+1} = F_t(\xi_t, a_t), \tag{1}$$

where $\delta_t$ is the policy that consumes the signal and produces an action $a_t$, and $F_t$ updates the state after the action is executed. This perspective makes explicit that the object of study in this survey is not confidence estimation in isolation, but the way confidence participates in control.

In many systems, the raw score is not consumed directly. Instead it is transformed by calibration, normalization, aggregation, ranking, semantic clustering, or thresholding before the final decision is made. We therefore write

$$\tilde{\kappa}_t = T_t(\kappa_t), \qquad a_t \sim \delta_t(\cdot \mid \xi_t, U_t, \tilde{\kappa}_t), \tag{2}$$

where $T_t$ denotes an optional preprocessing map. Thresholding is thus only one special case of a broader pattern: confidence may be used to select, weight, allocate, aggregate, abstain, escalate, backtrack, or otherwise alter the control flow of the system. Figure 2 summarizes this signal–transformation–policy–action template.

$$\underbrace{\xi_t}_{\text{decision state}} + \underbrace{U_t}_{\text{decision units}} \xrightarrow{\text{score } \kappa_t} \underbrace{\tilde{\kappa}_t = T_t(\kappa_t)}_{\text{optional transformation}} \xrightarrow{\text{policy } \delta_t} \underbrace{a_t}_{\text{action}} \xrightarrow{F_t} \underbrace{\xi_{t+1}}_{\text{updated state}}$$

Figure 2: Unified confidence-as-control template. A confidence signal is defined over decision units under a local decision state, optionally transformed, and then consumed by a policy that determines the next action.

This notation lets us characterize each method along three orthogonal axes. The first axis is the *source* of the signal: confidence may arise from the model itself through token probabilities, hidden states, or verbalized self-assessment; from sample-based behavior such as disagreement, self-consistency, or semantic entropy; from an auxiliary scorer such as a verifier, reward model, router, or judge; from external evidence or environment feedback such as retrieval signals and tool outcomes; or from peer agents in multi-agent systems. The second axis is the *unit or granularity* over which the signal is defined: token level, local content such as claims or chunks, item or candidate level, model/tool/agent level, step level, trajectory level, or full-episode level. The third axis is the *functional role* played by the signal in the downstream system: it may drive selection, weighting, resource allocation, control-flow decisions, aggregation, or serve directly as a learning signal.

With these axes in place, a method can be written compactly as

$$\kappa_t^{s,g}(u; \xi_t) \xrightarrow{T_t} \tilde{\kappa}_t \xrightarrow{\delta_t^r} a_t,$$

where $s$ indexes the source, $g$ indexes the unit or granularity, and $r$ indexes the functional role. When the context is clear, we omit these superscripts.

## 2.2 Scope: What Counts as Confidence Utilization

Because "confidence" admits many operational forms, we apply an explicit three-part inclusion test to every candidate method. A method is *in scope* when:

**C1** (*reliability semantics*) its signal $\kappa_t(u; \xi_t)$ is interpretable as a monotone estimate of a stated target property $\rho(u)$ concerning the reliability, correctness, groundedness, or expected success of the unit;

**C2** (*decision coupling*) the signal is consumed by a policy $\delta_t$ that changes system behavior, rather than being only reported, evaluated, or logged—thresholding a score into a system-facing flag or label that downstream logic acts on counts as decision coupling (the standard deployment of hallucination detectors), whereas reporting a score to a human reader or evaluating it against annotations does not; and

**C3** (*instance- and state-conditioned locality*) the signal is computed per decision unit and conditioned on the current decision state—at inference time the current query and generation state, at training time the current model and its beliefs about the unit—rather than being a unit-independent, corpus-level statistic.

Work that satisfies all three criteria with $\rho$ equal to the correctness or reliability of the unit itself is *core* to the survey, marked ● in all comparison tables. Work that satisfies C2 and C3 but whose target property is a real-valued quantity $\nu(u)$—expected utility, difficulty, data influence, generic quality, or reward, correlated with but not identical to reliability—is *adjacent*, marked ○; we retain such methods when they solve the same control problem as core methods and are needed to delineate the boundary. Estimation- and calibration-only work fails C2 and is cited as background; unit-independent corpus statistics and quality filters that condition on neither the learner nor the decision state fail C3 and are excluded. Training-time data selection thus remains in scope when its score is conditioned on the current model (e.g., instruction-following difficulty under the learner's own likelihoods), even though the filtering pass itself runs offline. This test also disambiguates

four senses in which the literature uses the word confidence: (i) confidence as *estimated correctness* (core), (ii) confidence as *expected utility* (adjacent), (iii) confidence as *verifier- or judge-derived quality* (core when the verified property is correctness, adjacent when it is generic quality or preference), and (iv) confidence as the *control value* $\tilde{\kappa}_t$ that a policy ultimately thresholds. Senses (i)–(iii) describe semantic families of $\kappa_t$; sense (iv) is the post-transformation quantity of Eq. 2 and exists in every method regardless of family. Two boundary rules keep the markers consistent across all comparison tables: a predicted *success or acceptance* probability of the unit—including acceptance by a reference model that operationally defines correctness, as in distillation and speculative verification—is core, whereas a predicted *difficulty, complexity, generic quality, preference strength, or expected benefit of an intervention* is adjacent, even when that prediction is computed from confidence-bearing features.

**Survey methodology.**  The corpus was assembled in three stages. First, we seeded from the estimation- and calibration-oriented surveys discussed in Appendix A and extracted all cited works whose contribution includes a downstream decision. Second, we searched the ACL Anthology, OpenReview (ICLR, ICML, NeurIPS, TMLR), and arXiv with query families covering signal vocabulary (confidence, uncertainty, calibration, semantic entropy, verifier, process reward) crossed with action vocabulary (selection, filtering, routing, cascading, deferral, abstention, adaptive retrieval, early stopping, backtracking, debate). Third, we snow-balled forward and backward citations of all included papers. Every candidate was scored against C1–C3; borderline cases were retained with the ○ marker and their boundary status is discussed in the text where they appear. The corpus emphasizes work from 2022 onward, when LLM-specific confidence signals became distinct from classical calibration, and covers papers appearing through early July 2026 (the corpus was refreshed once during revision). Borderline cases were adjudicated jointly by the authors against C1–C3, with disagreements resolved by discussion; the boundary rules stated above record the outcome of that adjudication. The corpus is curated under explicit criteria rather than being exhaustive: we prioritize methods that introduce a new signal–policy coupling over follow-up work that reuses an existing coupling, so absence from our comparison tables indicates absence from the curated corpus, not from the literature.

## 2.3   The Framework at Work

An organizing abstraction earns its place only if it produces observations that per-paper framings obscure. We highlight three uses of the source/unit/role decomposition, developed throughout the survey and summarized in Figure 3.

**Structural constraints.**  The axes are not freely combinable, and the constraints are informative. Decision timing restricts the source axis: a pre-call router (§5) must act before any candidate exists, so sample-agreement and answer-conditioned signals are structurally unavailable to it online (they reappear pre-call only when computed offline as router supervision, §5), which is why pre-call methods converge on auxiliary predictors while post-hoc deferral methods converge on self-signals—a systematic pattern of §5 that per-paper framings present as independent design choices. Similarly, granularity restricts the transformation axis: token-level signals must be aggregated before they can control answer-level actions, and §4 shows that the choice of aggregation (mean versus worst-group) is itself a load-bearing design decision rather than a detail.

**Cross-domain equivalences.**  The shared notation makes visible that several literatures maintain separate vocabularies for the same policy template. Cascade deferral (§5), risk-management abstention (§7), and retrieval triggering (§6) all instantiate the rule $a_t = \text{GET-HELP}$ if $\tilde{\kappa}_t < \tau$, differing only in what supplies the help: a stronger model, a human or a refusal, or external evidence. The consequence is practical, not cosmetic: because the families share a decision template, threshold-selection and calibration techniques developed for cascades are natural candidates for retrieval gating—after recalibration to the new signal distribution and cost structure—and the answer-conditioned versus answer-free timing distinction of §5 reappears in the when-to-retrieve literature of §6 as the same split (generate-then-check versus predict-before-generating) under different names. Likewise, adaptive-consistency stopping (§4) and cascade stopping rules instantiate the same optimal-stopping template over different unit sequences—samples from one model versus stages of a model portfolio—with correspondingly different signal distributions and costs.

|  | Selection | Weighting | Allocation | Control flow | Aggregation | Learning signal |
|---|---|---|---|---|---|---|
| **Model-internal** (logits, states, verbalized) | T I R K | T I | I | I M R K A | I | T K |
| **Sample agreement** (consistency, sem. entropy) | T I | I | I | M R K | I | T K |
| **Auxiliary scorer** (verifier, judge, RM, router) | T I A | T | I A | M R K A | *none in corpus* | T A |
| **External evidence** (retrieval, tools, env.) | R | *none in corpus* | *none in corpus* | R A | *none in corpus* | *none in corpus* |
| **Peer agents** (communicated confidence) | *none in corpus* | *none in corpus* | *none in corpus* | A | A | *none in corpus* |

Figure 3: The populated source × role design space, aggregated over the comparison tables of §3–§8. Cell labels indicate the lifecycle domains that instantiate each combination (T = training, I = inference, M = model selection, R = RAG, K = risk, A = agentic); shading scales with the number of domains populating a cell. Cells marked "none in corpus" are combinations not represented in our curated corpus (§2.2), not claims that no such work exists anywhere. The matrix exposes structure that individual sections cannot: sample-agreement signals are absent from pre-call decisions for timing reasons (§2.3), reappearing pre-call only where agreement is computed offline as router supervision (§5); external-evidence and peer signals are consumed almost exclusively as control-flow and aggregation inputs, and almost never as learning signals; and the aggregation role is concentrated at inference and agentic stages. The sparsely populated cells—external evidence as a training-time learning signal and peer confidence outside multi-agent deliberation—are, in our reading, open design space rather than infeasible combinations; the unit axis, which this matrix does not show, adds a third gap visible in the comparison tables: token-granularity control in model selection beyond speculative hand-off.

**Contradictions exposed.** Writing methods in a common form also surfaces oppositions that would otherwise read as unrelated results. The same comparison $\tilde{\kappa}_t < \tau$ triggers *opposite* actions at different lifecycle stages: in the data- and token-selection subfamily of training, low *self*-confidence *attracts* gradient mass because it marks learnable units (§3)—teacher- and reward-side training methods invert this polarity, trusting high-confidence supervision and damping uncertain rewards—while at deployment time low confidence *repels* commitment because it marks risk (§7); any system that tunes selection and deployment with one self-signal inherits this tension. Within a single stage, the transformation step $T_t$ is the site of a second conflict: calibration maps that improve pointwise reliability metrics can strictly worsen conformal set efficiency (§7), so "calibrate first, then control" is not a safe default. And across sources, verbalized self-confidence is simultaneously among the better-calibrated signals for RLHF answering and among the worse ones inside evaluator pipelines (§7), which the framework attributes to a change in the target property $\rho$—answer correctness versus judgment correctness—rather than to an inconsistency in the literature.

**Lifecycle specialization.** This abstraction specializes naturally across the lifecycle considered in this survey. In training, the decision unit is often a training example, preference pair, or token, and confidence governs filtering, reweighting, or reward assignment. In inference, it is typically a candidate response, partial trace, or reasoning step, and confidence governs selection, stopping, continuation, or revision. In routing and cascading, the unit is usually a model or route and the action is to choose, escalate, or defer. In retrieval-augmented generation, the relevant units include retrieval actions, chunks, and context sets, while the action may be to trigger retrieval, rerank evidence, filter passages, or halt. In risk management, the unit is often an answer, claim, or prediction set and confidence determines abstention, deferment, diagnosis, or coverage control. In agentic systems, the units may include tool invocations, branches, trajectories, or agent votes, and confidence may trigger search expansion, pruning, backtracking, replanning, or aggregation. The purpose of this framework is not to erase these differences, but to express them in a common language that makes the design space explicit—and, as Figure 3 shows, partially empty.

**Historical roots.** Confidence-as-control predates LLMs, and the framework deliberately echoes that lineage. The rejection option in classification (Chow, 1970) and its modern treatment as selective prediction (El-Yaniv & Wiener, 2010; Geifman & El-Yaniv, 2017) correspond to the abstention role at the answer unit; learning-to-defer (Madras et al., 2018; Mozannar & Sontag, 2020) anticipates cascade deferral with an explicit downstream expert; classifier cascades (Viola & Jones, 2001) are the direct ancestors of model cascading in §5; sequential analysis and optimal stopping (Wald, 1945) underlie adaptive-sampling rules in §4; uncertainty-driven acquisition in active learning (Settles, 2009) prefigures confidence-aware data selection in §3; and bandit and POMDP formulations (Lattimore & Szepesvári, 2020; Kaelbling et al., 1998) supply the decision-theoretic language for routing under partial observability, which §5 encounters when deferral policies must act on noisy self-verification. We point to these connections where they sharpen the analysis, and note that several open challenges of §9—notably decision-aware evaluation—have well-developed analogues in this older literature that the LLM community has only begun to import.

## 3 Confidence-Aware Training

Training is the earliest point in the LLM lifecycle where confidence changes what the model learns rather than what it outputs. In the notation of §2, the decision state $\xi_t$ now contains the current parameters, the minibatch, and any auxiliary teacher, reward, or judge models, while the decision units $u \in U_t$ may be full training examples, preference pairs, individual tokens, or complete rollouts. The resulting action is therefore training-specific: retain or discard a unit, assign it a loss weight, choose which part of a teacher signal to imitate, scale an RL reward, or teach the model to abstain. A central complication is that the semantics of the score vary substantially across papers. Some methods rely on self-confidence or uncertainty from the policy itself; others act on teacher confidence, reward-model uncertainty, self-reflection scores, or broader difficulty and utility proxies. The common structure is still the same: a reliability-relevant score over training units determines how gradient mass is allocated. Table 1 condenses these training-stage methods by signal source, unit, and update role.

### 3.1 Confidence-Aware Data Selection

Training-time confidence first appears as a curation signal. A large cluster of papers interprets low confidence, or a closely related difficulty score, as evidence of learning value. Li et al. (2024c) define instruction-following difficulty from the gap between answer likelihood with and without the instruction, and use that score to retain examples that are challenging in a model-specific way. Li et al. (2024b) show that this ranking transfers surprisingly well from weak to strong models, so a much smaller proxy model can perform the expensive filtering step. Han et al. (2025) combine uncertainty with graph-based influence, favoring examples that are both hard and structurally representative. In a nearby active-learning setting, Muldrew et al. (2024) use predictive entropy and preference certainty to decide which completion pairs deserve expensive preference labels. Across these methods, low confidence does not mean bad data; it often means that a sample lies near the model's current frontier of learnability.

Other data-selection methods treat consistency or judged quality, rather than raw difficulty, as the more useful signal. Liu et al. (2024a) rank instruction data using uncertainty-aware self-reflection across score tokens, paraphrased prompts, and multiple model scales, thereby rewarding examples whose quality assessment is both strong and stable. Chen & Mueller (2024) similarly use a confidence-bearing evaluator to filter noisy supervision and to rewrite targets only when the correction itself is judged reliable. Sachdeva et al. (2026) move the same logic to an external judge model, using the probability of a positive usefulness judgment as a quality score. These papers are closer to confidence-guided quality assessment than to pure difficulty filtering, because the signal is interpreted as expected usefulness or trustworthiness rather than mere challenge. Among them, Table 1 marks CLEAR core because its evaluator confidence estimates the correctness of the retained or rewritten target, while SelectIT and Ask-LLM score generic quality or usefulness and are marked adjacent under C1–C3.

The neighboring curation literature is important context, but it should not be conflated with confidence-aware training. LIMA (Zhou et al., 2023) is a quality-first human curation baseline rather than a confidence method; Abbas et al. (2023) prune redundancy through semantic similarity rather than uncertainty; Zhang

et al. (2025b) score examples by estimated holdout-loss impact; and Pan et al. (2026) use attribution to identify unsafe training data. Pang et al. (2025) further shift the decision unit below the sample level by removing low-value tokens inside otherwise useful examples, but its signal is closer to token influence than to confidence. We retain these works as nearby baselines because they solve the same control problem—which data should shape the model—even when the scoring semantics are not confidence in the strict sense.

### 3.2 Confidence-Aware Fine-Tuning, Distillation, and Abstention Tuning

Once the data has been chosen, confidence can act directly on the update rule. Krishnan et al. (2024) provide the clearest policy-side example: they augment causal language modeling with an uncertainty-aware objective that encourages low uncertainty on correct tokens and high uncertainty on incorrect tokens, with the explicit aim of making downstream uncertainty signals more usable for hallucination detection and selective generation. Li et al. (2025b) use contextual uncertainty differently. Its two-stage procedure first teaches the model to recognize when the provided evidence is insufficient, and then separately teaches compliance with abstention instructions, so that uncertainty becomes an answer-versus-refuse decision rather than only a calibration statistic. By contrast, Rahmati et al. (2025) mainly improve uncertainty estimation under parameter-efficient fine-tuning through contextual stochastic adapters; since the estimate is never consumed by a training decision, their method fails C2 and we cite it as background rather than marking it in Table 1.

Teacher-student settings make the control role even more explicit. Huang et al. (2025a) distill only where the teacher's propose-and-verify signal is trustworthy, so teacher confidence determines which tokens receive supervision. Zhong et al. (2024) likewise use teacher token uncertainty to split easy and hard tokens and to vary the distillation mode accordingly. In both cases, the confidence source is external to the student policy: the score decides where imitation should be sharp, softened, or withheld. Other training-control papers occupy the boundary of the category. Li et al. (2024a) use student-specific difficulty and compatibility statistics to select teacher-refined data, and Pang et al. (2025) use token-level influence to prune within-example supervision. These methods reinforce the broader point that training-time control often lives at finer granularities than whole samples, even when the operative signal is only confidence-adjacent.

### 3.3 Confidence-Aware Preference Optimization and RL

Confidence becomes most explicit in preference optimization and RL, where it can enter as a reward, a penalty, or a gating signal over updates. Several DPO-family baselines already optimize probability-derived surrogates: DPO uses policy-relative log-probability ratios (Rafailov et al., 2023), SimPO replaces the reference-based reward with average log probability (Meng et al., 2024), and $\beta$-DPO adapts optimization strength across batches (Wu et al., 2024). These methods are important background because they show that likelihood-derived quantities already function as training signals. However, later work makes the confidence role explicit rather than implicit.

On the policy side, Yoon et al. (2025) select low-confidence tokens as the sites where preference optimization should act, based on the observation that these tokens carry larger alignment gradients. Pokharel et al. (2025) modulate multilingual preference updates with a relative reward margin, treating stronger pairwise preference gaps as more trustworthy supervision. Lu et al. (2025) use local low-confidence points inside a reasoning trace to decide where to split, branch, and construct preference pairs. Du et al. (2025) turn final-answer confidence into a reward proxy for close-ended reasoning and then reuse confidence gaps to build stronger DPO data. In sequence-level RL, Li et al. (2025c) sharpen the policy around its own high-probability responses, Prabhudesai et al. (2026) use negative entropy as an intrinsic reward, Zhou et al. (2025a) reweight trajectories by sequence confidence and problem difficulty, and Liu et al. (2025b) jointly optimize reasoning accuracy and sequence-level calibration so that confidence remains useful after post-training. Taken together, these papers show that confidence can decide not only which output is preferred, but also where along a response or trajectory learning pressure should be concentrated.

A second line of work places the uncertainty in the supervision signal rather than in the policy output itself. Zhai et al. (2024) penalize rewards with high ensemble uncertainty so the policy does not overoptimize dubious reward spikes, while Banerjee & Gopalan (2024) derive a variance-aware conservative policy objective from the same intuition. Leng et al. (2025) locate the problem in the reward pipeline itself, showing that

| Method | Source | Signal | Unit | Training Action |
|---|---|---|---|---|
| **Data curation / fine-tuning / distillation** | | | | |
| ○ Cherry (Li et al., 2024c) | self | IFD / instruction-conditioned likelihood gap | example | select SFT examples |
| ○ Superfiltering (Li et al., 2024b) | self | weak-model IFD / difficulty ranking | example | proxy-filter examples for strong model |
| ○ UniMax (Han et al., 2025) | hybrid | uncertainty plus graph influence | example | select representative hard examples |
| ● Active-Pref (Muldrew et al., 2024) | hybrid | predictive entropy plus preference certainty | pair | acquire preference labels |
| ● SelectIT (Liu et al., 2024a) | hybrid | self-reflection consistency across tokens / prompts / models | example | select instruction data |
| ● CLEAR (Chen & Mueller, 2024) | hybrid | BSDetector confidence from agreement and self-reflection | example | filter and rectify supervision |
| ○ Ask-LLM (Sachdeva et al., 2026) | auxiliary | judge usefulness probability $P(\text{YES})$ | example | select pretraining data |
| ● UA-CLM (Krishnan et al., 2024) | self | token entropy / white-box uncertainty | token | reweight CLM loss for calibration |
| ● US-Tuning (Li et al., 2025b) | self | context-sufficiency uncertainty | question-context pair | tune answer-versus-abstain behavior |
| ● SelecTKD (Huang et al., 2025a) | auxiliary | teacher propose-verify acceptance confidence | token | distill only trusted tokens |
| ● ATKD (Zhong et al., 2024) | auxiliary | teacher token uncertainty / difficulty coefficient | token | switch distillation mode by token |
| ○ SRD (Li et al., 2024a) | self | IFD plus r-IFD difficulty / compatibility | example | select teacher-refined data |
| **Preference optimization / RL** | | | | |
| ○ DPO (Rafailov et al., 2023) | self | policy-relative log-probability ratio | pair | preference optimization baseline |
| ○ SimPO (Meng et al., 2024) | self | average log probability | pair | reference-free preference optimization baseline |
| ○ $\beta$-DPO (Wu et al., 2024) | self | batch informativeness / reward discrepancy | batch / pair | adapt optimization strength |
| ● ConfPO (Yoon et al., 2025) | self | below-average token probability | token | select alignment-critical tokens |
| ● CAPO (Pokharel et al., 2025) | hybrid | relative reward margin as pair confidence | pair | scale preference loss by strength |
| ● CGPO (Lu et al., 2025) | hybrid | token confidence + RM score | step | branch traces and build pairs |
| ● CRew (Du et al., 2025) | self | final-answer token confidence | answer / response | reward proxy and pair construction |
| ● RLSC (Li et al., 2025c) | self | old-policy response probability | response | self-confidence reward shaping |
| ● RENT (Prabhudesai et al., 2026) | self | negative entropy on late-response tokens | late tokens / trajectory | intrinsic RL reward |
| ● CoDaPO (Zhou et al., 2025a) | hybrid | sequence confidence plus problem difficulty | trajectory | reweight RL advantages |
| ● C2GSPG (Liu et al., 2025b) | self | normalized sequence probability | trajectory | calibration-aware sequence RL |
| ● UP-RLHF (Zhai et al., 2024) | auxiliary | reward-model ensemble variance | response | penalize uncertain rewards |
| ● UA-RLHF (Banerjee & Gopalan, 2024) | auxiliary | reward-model ensemble variance | response | conservative variance-aware RLHF |
| ● Taming-OC (Leng et al., 2025) | hybrid | verbalized confidence in reward calibration | response | calibrate reward model / reward score |
| ● BCHRL (Wu et al., 2025b) | self | verbalized confidence or critic value | response / claim | train risk-sensitive abstention policy |
| ● Explicit-Uncertainty RL (Guo et al., 2026b) | self | Brier-rewarded verbal confidence + marker | response / step | train calibrated explicit channel |
| ● TruthRL (Wei et al., 2026) | hybrid | ternary correct/abstain/hallucinate reward | response | train truthful abstention |

Table 1: Training-stage confidence-utilization methods, organized by signal source, signal form, decision unit, and training action. The table merges data curation, fine-tuning, distillation, preference optimization, and RL into one grouped comparison. Methods are marked core (●) or adjacent (○) according to the inclusion criteria C1–C3 of §2.2.

RLHF reward models prefer overly confident responses and then correcting this bias either in reward-model training or in reward calculation during PPO. Wu et al. (2025b) go one step further and train the model to answer only when its estimated correctness exceeds a user-specified risk tolerance, thereby making confidence operationally identical to an abstention policy. Recent evidence also clarifies *which* confidence channel RL helps or harms. Policy-gradient and preference objectives sharpen the answer distribution and thereby degrade *implicit* token-probability calibration—self-distilled SFT preserves it at comparable accuracy (ECE 0.034 versus 0.135 for GRPO and 0.117 for DPO in the controlled comparison of Xie et al., 2026)—yet the same GRPO machinery can install a calibrated *explicit* uncertainty channel when the reward targets it directly: Brier-style rewards on verbalized confidence cut ECE from 0.383 to 0.049 without hurting accuracy (Guo et al., 2026b), and a ternary correct/abstain/hallucinate reward makes abstention a reward-optimal action where binary rewards conflate it with error (Wei et al., 2026). Training-time confidence design is therefore channel-specific: RL damages the implicit signal even as it can shape an explicit one (§7.4). Not every token-level alignment refinement is confidence-aware, however: Liu et al. (2025a), for example, reweight tokens by estimated importance rather than reliability. This distinction is useful because it separates confidence utilization from the broader family of fine-grained optimization heuristics.

**Discussion.** Across these families, training uses confidence to allocate gradient mass. The same arithmetic operation can therefore mean very different things depending on the source of the signal. Low self-confidence may mark high-learning-value examples or tokens, as in Cherry, ConfPO, or CGPO. High external confidence may mark trustworthy supervision, as in SelecTKD, CLEAR, or Ask-LLM. High reward uncertainty, by contrast, often suppresses optimization, as in UP-RLHF and UA-RLHF. This is precisely why the source/unit/role decomposition from §2 is most useful in the training setting: without it, difficulty, quality, preference strength, teacher reliability, and reward uncertainty all collapse into a single overloaded notion of confidence. The main lesson of this section is therefore structural rather than metric-specific. During training, confidence matters because it determines where learning should occur, where it should be damped, and when the model should be explicitly taught to abstain; in later sections, the same family of signals will shift from controlling gradient allocation to controlling system behavior at deployment time.

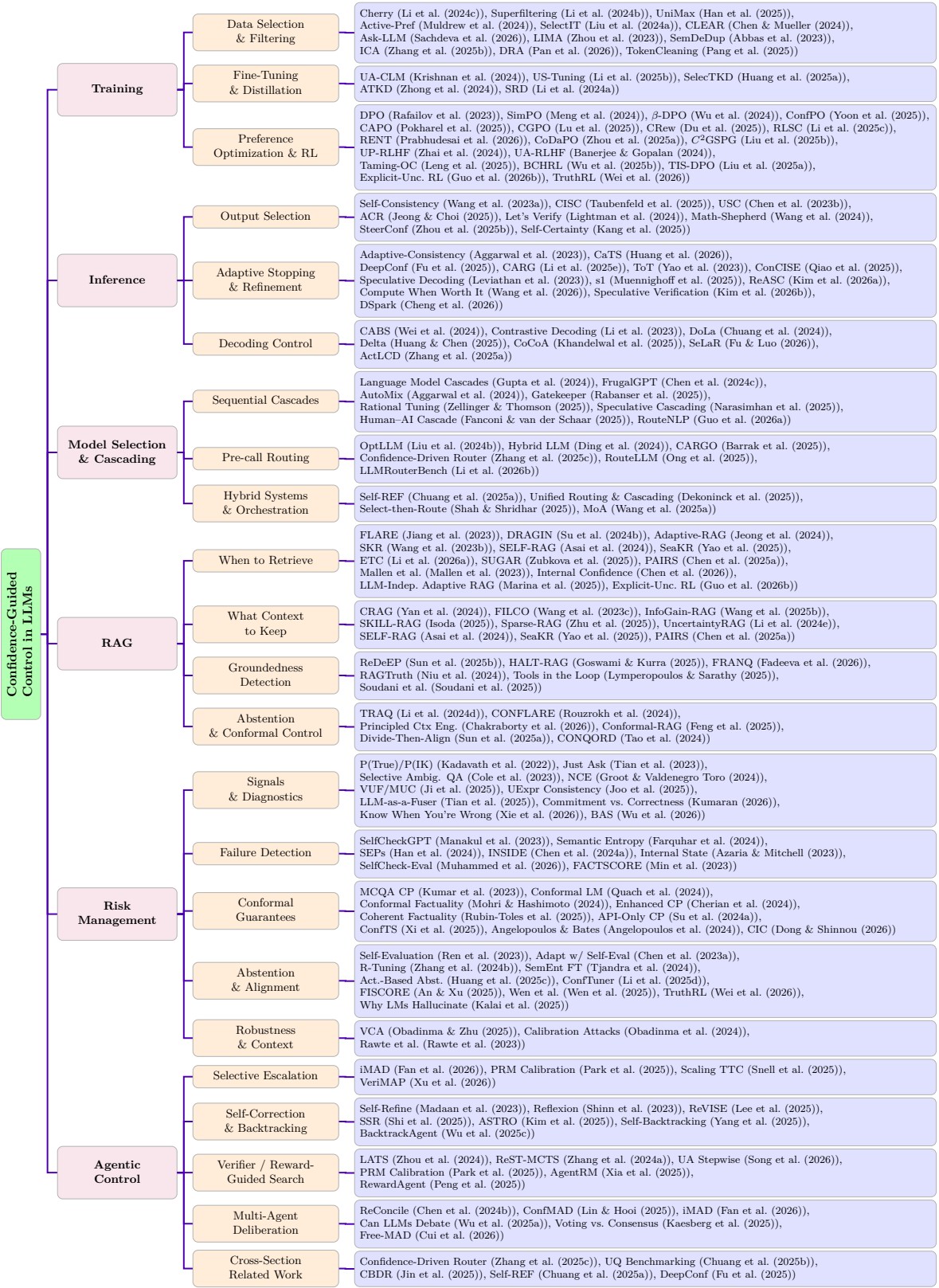

Figure 4: Taxonomy of confidence utilization in LLMs across six domains. Leaves list representative core, adjacent, and contextual works with author–year citations; the comparison tables in §3–§8 give the complete core/adjacent inventories.

# 4 Confidence-Driven Inference

Inference is where confidence becomes an online control variable. In the notation of §2, the decision state $\xi_t$ now contains the prompt, partial generations, sampled candidates, any verifier or judge outputs, and the remaining inference budget, while the decision units $u \in U_t$ may be complete candidate responses, partial reasoning states, token groups, or next-token alternatives. The resulting action therefore depends on where the signal is injected: at the candidate level it selects or aggregates complete outputs, at the state level it decides whether to continue, stop, refine, or backtrack, and at the token level it reshapes the predictive distribution itself. The literature in this section spans both explicit confidence signals, such as answer agreement or elicited confidence, and broader reliability surrogates, such as verifier scores, local log-probability drops, or disagreement between predictive views. We organize the section accordingly. Table 2 provides a compact view of the main inference-stage methods and the control decisions they support.

## 4.1 Confidence-Driven Output Selection

The most direct use of inference-time confidence acts on a fixed candidate set. Wang et al. (2023a) establish the basic template with self-consistency: sample diverse reasoning paths, extract their final answers, and use answer agreement as an implicit confidence signal for final selection. Later work asks how to make this agreement signal more informative. Taubenfeld et al. (2025) augment self-consistency with path-level confidence scores and show that the decisive property is not global calibration, but *within-question discrimination*: confidence must separate stronger and weaker responses to the same question if it is to improve multi-sample selection. Their CISC procedure therefore normalizes candidate confidence within each question and performs confidence-weighted voting, with $P(\text{True})$ emerging as especially effective for this use case. In the same spirit, Chen et al. (2023b) remove the requirement that answers be cleanly extractable. Universal Self-Consistency asks the model itself to choose the most consistent response among sampled candidates, so the confidence signal is comparative rather than scalar: the model acts as a consistency-based selector over an answer set rather than producing an independent score for each answer.

This family also supports more selective use of confidence. Jeong & Choi (2025) begin from an ordinary self-consistency distribution and trigger an additional re-scoring stage only when that distribution is too flat to be trusted. The first-stage candidate frequencies define an ambiguity signal; only ambiguous cases are passed to a second-stage multiple-choice selection step, whose outputs are then combined with the original self-consistency scores. Confidence here is therefore used twice: first to detect when majority voting is unreliable, and then to support final answer choice among the surviving candidates.

An orthogonal line of work replaces self-derived confidence with externally trained verifier signals. Lightman et al. (2024) show that process reward models trained with step-level supervision are stronger search-time selectors than outcome-only reward models, precisely because they can discriminate errors that occur inside otherwise plausible full solutions. Wang et al. (2024) reduce the annotation burden by constructing process labels automatically, but the inference role is similar: a step-level verifier scores candidate solutions and the aggregate verifier score determines which completed solution should survive. These papers belong in the inference section because the verifier signal governs selection at test time, even though the signal itself is produced by a separately trained auxiliary model.

Finally, some methods are better understood as confidence-elicitation primitives that support downstream selection. Zhou et al. (2025b) use prompt steering to obtain multiple verbalized confidence views of the same answer, combine mean confidence with answer consistency and confidence consistency, and then use the resulting calibrated score for answer selection. This makes SteerConf highly relevant to inference, but its main contribution is black-box confidence elicitation rather than multi-candidate reasoning control in the narrower self-consistency sense.

A related training-free signal sharpens the top of the voting family. Kang et al. (2025) score each candidate by *self-certainty*—mean token-level KL divergence from the uniform distribution—and fuse the resulting ranking with majority voting through Borda weights. The rank-based fusion edges plain self-consistency at large sample counts (56.51 versus 56.15 average over their math suite at $N{=}64$; MATH 64.10 versus 63.40), keeps improving to $N{=}64$ where perplexity- and average-log-probability-family signals plateau or decline

after $N\approx16$, and—being length-invariant—extends confidence-weighted selection to code and open-ended outputs where exact-match voting is undefined.

Within the controlled comparisons available, these papers exhibit a recurring empirical ordering for candidate selection: trained process reward models > confidence-weighted voting > plain majority voting > raw sequence probability. Where step-level supervision and a strong verifier are available, PRMs dominate at scale: at best-of-1860 on a 500-problem MATH held-out set, the PRM of Lightman et al. (2024) reaches 78.2% against 72.4% for the outcome-supervised model and 69.6% for majority voting, and the gap *widens* with more samples while majority voting plateaus near $N=100$. Where no verifier exists, confidence-weighted voting is the strong training-free default: CISC with $P(\text{True})$ weights matches self-consistency's 30-sample accuracy with a 46% smaller budget at ten samples (vanilla self-consistency needs 18.6 samples to match CISC at ten). Raw sequence probability sits at the bottom—Wang et al. (2023a) themselves report that unnormalized probability-weighted voting collapses GSM8K selection from 74.4 to 59.9 because sampled paths receive nearly indistinguishable likelihoods. The CISC analysis explains why the ordering looks this way: the property that predicts selection gains is within-question discrimination, not calibration. Verbal binary confidence attains the best calibration in their comparison (temperature-scaled ECE 0.005) yet is the weakest voting weight (10% cost reduction), while $P(\text{True})$ is only moderately calibrated (0.030) but has the highest within-question discrimination (62.3%) and delivers the largest gains—standard calibration metrics rank signals incorrectly for this control decision. Even the PRM-versus-ORM conclusion is not unconditional. Lightman et al. (2024) establish the gap with human step labels, GPT-4-scale generators, and MATH; Wang et al. (2024) reproduce the PRM > ORM > majority ordering with automatic labels (LLaMA2-70B GSM8K: 93.2 vs. 91.8 vs. 88.0) but observe that the gap shrinks on GSM8K, that combining the verifier with self-consistency helps MATH (45.2 vs. 44.5) yet *hurts* GSM8K (92.4 vs. 93.2), and that their automatically labeled PRM outperforms a PRM trained on human-labeled PRM800K data when scoring their own generators' outputs. Verifier conclusions are therefore generator-, scale-, and dataset-conditional rather than settled facts about supervision form.

When does this family work? Agreement-based selection presupposes a fixed, exact-matchable answer space and a sampling budget, and its LLM-judged relaxation in USC—which matches counted self-consistency where both apply (GSM8K 90.2 vs. 90.4)—degrades as the candidate set grows, with GSM8K accuracy *decreasing* at sixteen samples while counted voting improves monotonically. Confidence-weighted variants additionally require token-probability access and, per CISC's ablations, temperature-scaled softmax normalization; raw confidence scores transfer poorly across models without it.

## 4.2 Adaptive Stopping, Refinement, and Search

Confidence can also act before the candidate set is complete. In this regime, the relevant units are partial answer distributions, intermediate reasoning segments, or search states, and the action is to continue, stop, refine, or revisit the current trajectory. Aggarwal et al. (2023) provide the cleanest agreement-based example: Adaptive-Consistency samples one candidate at a time and stops when the estimated probability that the current majority answer will remain dominant is sufficiently high. Confidence is therefore not used to rank completed outputs, but to determine whether additional sampling is still worth the compute. Huang et al. (2026) push the same logic further by first distilling self-consistency-derived confidence into the model itself and then using the calibrated score for weighted voting, early stopping in Best-of-$N$, and adaptive self-consistency. The paper thus straddles post-training and inference, but its operational contribution is squarely inference-time budget control.

Fu et al. (2025) show that the relevant signal need not live at the full-trace level. Their DeepConf framework derives confidence from internal token probabilities and finds that local low-confidence regions, especially the least-confident token groups, are more diagnostic than whole-trace averages. This signal then supports two distinct actions: offline confidence-weighted filtering or voting over completed traces, and online early stopping when a running trace enters a sufficiently weak local segment. Li et al. (2025e) study a different inference setting—sequential interactions rather than one-shot reasoning—but the control logic is similar. Their CARG framework extracts response confidence from answer-token probabilities and uses it to decide whether a model should maintain its current answer or reconsider it under later follow-up pressure. Confidence here functions as a persistence-versus-revision signal over turns.

More broadly, search-style reasoning methods also employ confidence-like state evaluation. Yao et al. (2023) use self-value and self-vote prompts to score partial thought states, then use those scores to expand promising branches and prune weak ones. Under the broad operational definition adopted in this survey, these state-evaluation scores function as confidence over incomplete reasoning states rather than as confidence over completed answers. ConCISE (Qiao et al., 2025) is a boundary case between inference and training: it uses confidence injection and stopping heuristics to suppress redundant reflection and construct concise reasoning traces, but much of the eventual gain comes from fine-tuning on the compressed data rather than from leaving the control rule purely at inference time.

This subsection also clarifies an important boundary of scope. Nearby test-time scaling papers such as speculative decoding (Leviathan et al., 2023) and budget forcing in *s1* (Muennighoff et al., 2025) adapt compute at inference time, but they do not do so on the basis of a reliability signal in the sense of this survey. We therefore treat them as adjacent compute-control baselines rather than as core confidence-utilization methods. That boundary has recently begun to move, however, because the *verification step* of speculative decoding is itself becoming confidence-scheduled. Kim et al. (2026b) attach a draft-sized companion model whose post-drafting agreement signals (distribution overlap and pseudo-acceptance) recover 30–40% of the entropy of speculation outcomes and support goodput-optimal verification-length scheduling worth up to $1.9\times$ over vanilla speculative decoding at large batch—exactly the regime where draft-internal confidence signals fail. Cheng et al. (2026) push this to production scale: a trained per-position survival head schedules how many drafted tokens each request submits to batched verification, delivering a 51% aggregate throughput gain at fixed per-user latency in DeepSeek-V4 serving. Notably, because DSpark's scheduler compares *absolute* survival probabilities across requests, ranking quality alone (ROC-AUC 0.81–0.90) was insufficient and the head had to be explicitly calibrated (sequential temperature scaling cuts its ECE from 3–8% to about 1%)—a clean counterpoint to the CISC finding above that selection needs discrimination rather than calibration: the control role, not the signal, determines which metric binds (§9).

The budget-control evidence, read across papers, traces a clear trajectory from agreement-only stopping toward internal-confidence stopping. Adaptive-Consistency's count-based criterion already recovers most of the waste in fixed-budget sampling—an average $3.3\times$ sample reduction on mathematical reasoning (40 down to 13.8 samples at $-0.1$ accuracy) and up to $7.9\times$ on Boolean Expressions with no loss—and, as noted in §2, its stop-or-continue rule instantiates the same optimal-stopping template as cascade deferral at the model level, under different signal distributions and costs. But count-only schemes saturate, and the successor generation makes the counting itself confidence-weighted: ReASC (Kim et al., 2026a) keeps Adaptive-Consistency's Beta-posterior stopping rule but updates it with pseudo-counts weighted by DeepConf-style bottom-decile group confidence, adding a single-sample acceptance gate, and attains the best accuracy-per-TFLOP in all twenty model–dataset cells tested, cutting compute by up to 70% against fixed self-consistency where count-based stopping saves less at equal accuracy. In the matched-budget comparison of Huang et al. (2026), count-based adaptive-stopping baselines likewise yield near-zero gains over self-consistency at budget 16, whereas the same rules re-weighted by calibrated confidence do better: calibrated confidence-weighted voting reaches self-consistency's 85.0% MathQA accuracy with 94.2% fewer samples, and the distilled confidence matches a dedicated external reward model at Best-of-16 (84.0 vs. 82.1 on MathQA)—evidence that trained verifiers are not strictly necessary at moderate budgets. DeepConf pushes the signal inside the trace and reports the strongest numbers in this family: online, GPT-OSS-120B on AIME25 reaches 97.9% with 84.7% fewer tokens than 512-sample voting (97.1%); offline, top-10% confidence filtering with weighted voting reaches 99.9% where 512-vote consensus saturates at 97.0%. DeepConf's ablations also echo the aggregation lesson of §2: lowest-group and bottom-decile statistics outperform whole-trace mean confidence, so the aggregation choice, not just the signal, carries the method. The trajectory's endpoint has also moved from *efficient* to *certified*: Wang et al. (2026) select a dual-threshold exit rule—an upper "confident-stop" and a scheduled lower "hopeless-stop" that terminates likely-unsolvable instances—by upper-confidence-bound risk control, so that wrong-early-exit risk stays below a user-specified level with finite-sample validity from as few as fifty calibration examples, converting the heuristic threshold tuning of the methods above into certified budget control and formally connecting adaptive stopping to the abstention machinery of §7. Two tensions temper this trajectory. First, sequential stopping trades throughput for latency—Taubenfeld et al. (2025) explicitly criticize this family on those grounds, since parallel confidence weighting preserves latency while adaptive schemes serialize sampling. Second, internal-confidence signals inherit systematic overconfidence: Fu et al.

(2025) report that aggressive top-10% filtering fails when the model is confidently wrong, which makes the filtering ratio dataset-dependent.

These methods work when answers are exact-matchable, additional samples are cheap to score, and the modal or high-confidence trace is usually right; Aggarwal et al. (2023) state plainly that their criterion can trigger on unstable majorities and is expected to fail wherever self-consistency fails, while DeepConf additionally requires log-probability access and representative warmup traces to set its stopping threshold.

### 4.3 Confidence-Shaped Decoding Control

At the finest scale, the decision unit is the next-token distribution itself. Here many methods do not estimate an explicit scalar confidence over a completed answer; instead they use disagreement between predictive views as an implicit token-level surrogate for reliability. Li et al. (2023) provide the canonical example. Contrastive decoding compares an expert model with a weaker amateur model and prefers tokens whose probability under the expert is high relative to the amateur, subject to a plausibility constraint that prevents the contrastive objective from selecting implausible tokens. The signal is therefore an expert–amateur disagreement score that directly reweights token choices. Chuang et al. (2024) internalize this contrast within a single network by comparing late and early layers, using layer disagreement as a token-level control signal for factual decoding. Zhang et al. (2025a) make this control adaptive: rather than applying layer contrast at every token, ActLCD learns a policy for when contrastive intervention is actually needed.

Other methods make the underlying confidence interpretation more explicit. Huang & Chen (2025) use masked-context contrast to identify tokens whose support is fragile under perturbations of the provided context, so the signal should be understood as contextual fragility rather than general answer correctness. Khandelwal et al. (2025) similarly frame token control as adaptive trust in context: they combine prior–context divergence, entropy gap, and contextual peakedness into a gating rule that determines how strongly decoding should follow the context-conditioned distribution rather than the model's prior distribution. The resulting signal is hybrid by construction, since it is defined by the relation between parametric and context-conditioned views rather than by either one alone. Fu & Luo (2026) extend this token-level control view to latent reasoning: normalized top-$k$ entropy gates whether decoding should preserve the ordinary discrete token embedding at high-confidence steps or activate soft latent embeddings at low-confidence exploratory steps, with an entropy-aware contrastive regularizer used to keep multiple latent alternatives alive. Finally, Wei et al. (2024) show that the relevant inference unit need not be a token at all. Their CABS framework learns a hidden-state-based confidence model over generated sub-structures and then uses those local confidence scores to guide beam search in structured generation, underscoring that the unit axis from §2 remains important even inside a single inference stage.

This family invites the most caution about claims that travel beyond their evidence. It is worth stating precisely what Li et al. (2023) did and did not show: the original contrastive-decoding paper evaluates open-ended text *quality* only—human raters prefer its outputs to nucleus sampling by 2.6× and to typical decoding by 6.4× on coherence—and contains no reasoning experiments; the popular claim that contrastive decoding improves reasoning originates in later, modified variants of the method; Table 2 accordingly marks it adjacent, since the original target property is text quality rather than token correctness. The controlled head-to-head in Chuang et al. (2024) is the strongest within-family evidence on that question, and it is negative: with a 7B amateur, contrastive decoding drops GSM8K from 33.8 to 28.4 (LLaMA-33B) and from 16.7 to 9.1 (13B) and hurts StrategyQA at all model sizes (69.9 to 66.7 at 33B), while DoLa improves both (GSM8K 51.2 to 54.0 at 65B; StrategyQA 60.1 to 64.1 at 7B). Sweeping alternative amateurs narrows but does not close the gap (65B GSM8K: baseline 51.18, best contrastive-decoding amateur 47.08, DoLa 53.60). The diagnosis offered— a small amateur still possesses partial reasoning ability, so subtracting its distribution removes reasoning signal, whereas early layers do not—also explains why, in the controlled comparisons available so far, within-model layer contrast outperforms between-model contrast when answer correctness rather than text quality is the target metric. Both signals are nonetheless scale- and configuration-sensitive: the contrastive-decoding ablations show that removing the plausibility constraint is catastrophic (MAUVE drops to 0.01) and that an $n$-gram or same-size amateur fails, while DoLa collapses on GPT-2 Medium (FACTOR News 41.0 to 22.2) and its best layer bucket flips between short factual outputs and long chain-of-thought tasks. The

| Method | Source | Signal | Unit | Role | Access |
|---|---|---|---|---|---|
| **Output selection** | | | | | |
| ●Self-Consistency (Wang et al., 2023a) | self | answer agreement | candidate answer | vote | MS |
| ●CISC (Taubenfeld et al., 2025) | self | $P$(True) / path confidence | path / candidate | weighted vote | MS |
| ●Self-Certainty (Kang et al., 2025) | self | KL-from-uniform + Borda rank fusion | candidate answer | weighted vote, select | WB, MS |
| ●Universal SC (Chen et al., 2023b) | self | comparative consistency judgment | candidate response | select | MS |
| ●ACR (Jeong & Choi, 2025) | self | SC frequency + ambiguity score | candidate answer | trigger, rescore | MS |
| ●PRM (Lightman et al., 2024) | auxiliary | process reward score | step / solution | rerank | AV, MS |
| ●Math-Shepherd (Wang et al., 2024) | auxiliary | PRM step score | step / solution | rerank | AV, MS |
| ○SteerConf (Zhou et al., 2025b) | self | steered verbal confidence | answer | calibrate, select | BB |
| **Adaptive stopping, refinement, and search** | | | | | |
| ●Adaptive-Consistency (Aggarwal et al., 2023) | self | majority stability | answer set | stop | MS |
| ●ReASC (Kim et al., 2026a) | self | bottom-decile group self-certainty | response / answer set | gate, weighted stop | WB, MS |
| ●Compute When Worth It (Wang et al., 2026) | self | risk-certified dual thresholds | reasoning trajectory | stop, abstain | WB |
| ●CaTS (Huang et al., 2026) | self | calibrated response confidence | candidate answer | vote, stop | MS, FT |
| ●DeepConf (Fu et al., 2025) | self | lowest-group logprob | token group / trace | filter, stop | WB, MS |
| ●Firm-or-Fickle (Li et al., 2025e) | self | answer-token probability | response / turn | maintain, revise | WB |
| ●ToT (Yao et al., 2023) | self | self-value / self-vote | thought state | expand, prune | MS |
| ○ConCISE (Qiao et al., 2025) | self | reflection confidence | step / trajectory | stop, compress | FT |
| **Confidence-shaped decoding control** | | | | | |
| ○Contrastive Decoding (Li et al., 2023) | hybrid | expert–amateur gap | token | reweight | 2M |
| ●DoLa (Chuang et al., 2024) | self | layer contrast | token | reweight | WB |
| ●Delta-CD (Huang & Chen, 2025) | self | masked-context contrast | token | reweight | WB |
| ●CoCoA (Khandelwal et al., 2025) | hybrid | prior–context conflict | token | blend, reweight | WB |
| ●SeLaR (Fu & Luo, 2026) | self | top-$k$ entropy | token / latent step | gate, regularize | WB |
| ●ActLCD (Zhang et al., 2025a) | self | learned contrast trigger | token | gate, reweight | WB, FT |
| ●CABS (Wei et al., 2024) | self | sub-structure confidence | sub-structure | beam rerank | WB, FT |
| ●Speculative Verification (Kim et al., 2026b) | auxiliary | companion-model draft agreement | token block | schedule verification | 2M, WB |
| ●DSpark (Cheng et al., 2026) | auxiliary | calibrated survival-probability head | token block | schedule verification | FT, WB |

Table 2: Inference-stage confidence-utilization methods, organized by signal source, signal form, decision unit, functional role, and operational access. Markers apply the inclusion criteria of §2: ● marks core methods whose signal estimates the correctness or reliability of the decision unit, while ○ marks adjacent methods (e.g., elicitation primitives, boundary cases whose gains partly come from training, or contrasts whose original target property is text quality) discussed for context.

*Access:* MS = multiple samples or search branches; WB = white-box logits or hidden states; AV = auxiliary verifier; BB = black-box prompting; 2M = second model at inference; FT = extra post-training or learned controller. Adjacent compute-control baselines such as speculative decoding and *s1* are discussed in the text but omitted here because they do not use a reliability signal in the sense of this survey.

compensation is cost: DoLa adds only 1.01–1.08× latency and at most 1.1% memory—the cheapest control action in this section, with relatively modest and configuration-sensitive gains.

By the papers' own account, these token-level interventions work best on factuality-style tasks with sufficiently large models when the sampling budget is essentially one; they cannot correct misinformation absorbed during pretraining, transfer poorly to small models, and require per-task validation of the contrast configuration.

**Discussion.** Across these families, inference shifts confidence from gradient allocation to online control. Candidate-level methods primarily aggregate or select among completed outputs; state-level methods determine whether reasoning should continue, stop, branch, or be revised; token-level methods reshape the predictive distribution directly. The same caution from §3 reappears, however: confidence is not semantically uniform across these papers. Agreement frequency, $P$(True), verbalized confidence, verifier scores, local log-probability drops, and contrastive disagreement are not interchangeable, even if they all influence the next inference action. Making these distinctions explicit is what separates core confidence-driven reasoning methods from adjacent compute-control or decoding heuristics. It also prepares the transition to routing and cascading, where the decision unit changes again—from candidate outputs within one model to model choices across a portfolio.

## 5 Confidence-Guided Model Selection

This section studies confidence-guided model selection in systems with multiple candidate models. Under the notation of Section 2, the decision state includes the query, the available model pool, the remaining budget or latency allowance, and, in some methods, a provisional answer or partial generation. The decision

units are therefore models, cascade stages, or token-level hand-off points, depending on when the system intervenes. The confidence signal may come from the small model itself, from an external router, from judge or verifier signals, or from uncertainty summaries such as semantic entropy. The downstream action is to select a model, defer to a stronger one, continue with the current model, or aggregate multiple outputs.

Relative to earlier sections, model selection introduces a particularly important timing distinction. Some methods make an ex-ante decision before any candidate model is run; others make a post-hoc deferral decision after a cheaper model has already produced an answer; and a smaller set intervenes mid-generation by handing off or verifying partial outputs. This timing axis determines both which confidence signals are available and what the signal means. Pre-call routing typically uses predicted suitability or expected quality, whereas post-call deferral is able to condition on an actual candidate answer and therefore use answer-conditioned uncertainty or verification scores. Table 3 organizes these routing and cascade methods by timing, signal source, and downstream action.

## 5.1 Confidence-Guided Selection Architectures

Deploying heterogeneous LLMs creates a persistent cost–quality tradeoff: stronger models are usually more reliable, but they are also slower or more expensive. Confidence-guided selection addresses this tradeoff by making the amount of computation conditional on the query and, in some cases, on a provisional answer. Earlier NLP work on model cascading already showed that instance-dependent escalation across classifiers can jointly improve efficiency and accuracy—matching a large model at 11–24% of its inference cost and sometimes exceeding its accuracy outright (Varshney & Baral, 2022); the recent LLM literature inherits this same control problem, but with richer confidence signals and sharper cost–quality tradeoffs. The literature in this section falls into three broad families: sequential cascades that defer uncertain cases to stronger models, pre-call routers that predict which model is most suitable before generation, and hybrid systems that combine both mechanisms.

**Sequential cascading: deferral as answer-conditioned escalation.** Sequential cascades run cheaper models first and defer only uncertain cases to stronger models. A core question in this setting is how to summarize the weaker model's uncertainty in a way that is useful for the deferral decision, and the classification-era answer does not survive the move to generation. In the classification cascades of Varshney & Baral (2022), maximum softmax probability is essentially optimal as a deferral signal. Gupta et al. (2024) show that its generative analogue—summed sequence log-probability—is strongly length-biased and can be *worse than random* deferral: on TriviaQA the Chow-Sum rule scores 13.1% below random on the deferral curve, and on MNLI it predicts whether deferral will pay with below-chance AUC-ROC (0.46), because long, repetitive outputs accumulate low sequence probability regardless of quality (output length correlates with BLEURT at −0.58 on their translation tasks). Their remedy treats token-level uncertainty as the primitive: quantiles over per-token log-probabilities, fed to a small learned router, recover gains of 10–20% over random where Chow-Sum fails, and adding intermediate decoder embeddings raises deferral-label AUC-ROC to 0.95. The correction is not universal—on TyDiQA-Swahili the raw sum remains the best signal—so even within one model family the right aggregation is dataset-dependent, echoing the aggregation lesson of §4.

Chen et al. (2024c) broaden the picture from one cascade rule to a larger cost-aware orchestration framework. FrugalGPT studies prompt adaptation, model approximation, and cascades under a unified cost–quality objective, learning both the model chain and per-stage thresholds for a DistilBERT answer scorer. Its headline economics remain a reference point: matching GPT-4 accuracy on HEADLINES at 1.7% of the cost ($0.6 versus $33.1), and *exceeding* GPT-4 by 1.5% accuracy at 80% lower cost at a fixed budget— possible because roughly 6–13% of GPT-4's errors are answered correctly by much cheaper models, so a cascade can exploit model diversity that a single model cannot. In the present survey it is best viewed as a foundational routing-and-cascade system: the control signal is a learned utility estimate rather than an explicitly calibrated confidence score, which is precisely where later work attacks it.

Aggarwal et al. (2024) are closer to a direct confidence-utilization view, and they supply the sharpest head-to-head in this family. AutoMix generates an initial answer with a cheaper model, then asks that same model to self-verify its output through a few-shot entailment prompt, estimating confidence from eight sampled verification verdicts. Because this signal is noisy, the routing policy is formulated as a POMDP over latent query difficulty rather than as a fixed threshold. Under their incremental-benefit-per-cost metric, AutoMix

beats FrugalGPT's trained router nearly everywhere—$\Delta_{\text{IBC}}$ 156.8 versus 16.8 on DIPLOMAT with Mistral-7B—while FrugalGPT turns *negative* on some datasets (COQA $-16.7$), meaning its learned scorer does worse than randomly interpolating between the two models. The out-of-distribution comparison is starker still: AutoMix retains $\Delta_{\text{IBC}}$ of 28–71 across model configurations where FrugalGPT collapses to 0–14, and AutoMix needs only about 50 training examples where learned routers need roughly a thousand. The lesson is about supervision, not architecture: a task-general self-verification prompt transfers where a dataset-specific scorer memorizes. The victory is conditional, however—by the paper's own analysis, gains disappear on QUALITY and QASPER, where verifier confidence stops correlating with correctness, and the method is explicitly restricted to context-grounded tasks where there is something to verify against.

Two later papers show that the deferral signal itself can be manufactured rather than merely read off. Rabanser et al. (2025) fine-tune the smaller model with a loss that sharpens softmax confidence on correct predictions and flattens it toward uniform on errors: correct-versus-incorrect separation rises from ROC-AUC 0.77–0.83 to as high as 0.94 across their vision and language suites, deferral performance improves by factors of 7–10× on ARC-e/ARC-c for a Gemma-2B→7B cascade, and the tuned signal beats the post-hoc learned router of Gupta et al. (2024) as well as prompting-based confidence baselines, which are unreliable across their experiments (consistent with Kadavath et al., 2022). The price is explicit: as the tuning knob shifts toward deferral quality, the small model "unlearns" hard examples and its standalone accuracy drops—confidence tuning converts model capacity into signal quality. Zellinger & Thomson (2025) leave the weights alone and instead treat threshold selection as a statistical problem: calibrate raw confidences by logistic regression (cutting ECE by 28.2% on average; their census finds GPT-4o-Mini reporting raw confidence of exactly 1.0 on 45.7% of MMLU questions, against 0.1% for Llama-405B), then fit a copula model of the joint confidence distribution across cascade stages and optimize all thresholds continuously. The payoff is regime-dependent in an instructive way: against high-resolution grid search and Bayesian optimization, their method wins mainly for cascades of length $k \geq 3$ (error–cost AUC $-4.3\%$ versus BO, $-7.2\%$ at $k{=}5$) and in low-data threshold tuning ($-10.2\%$ at $n \leq 30$); at $k{=}2$ the authors themselves recommend grid search. Their diagnostics also flag where the modeling breaks: the copula fit degrades exactly for cross-provider cascades (22.2% goodness-of-fit rejections, versus 1.7% within the Llama family)—the practically interesting heterogeneous case.

Two extensions push cascading beyond the two-model, fixed-threshold template. Fanconi & van der Schaar (2025) split the single deferral knob into two: Bayesian Platt scaling of the small model's $P(\text{True})$ yields a posterior whose *mean* (confidence) decides deferral to the larger model and whose *spread* (uncertainty) decides abstention to a human, with all thresholds learned online from human feedback on the abstained cases—the lowest cumulative regret on four of five datasets over 1,000 online steps, and a NeurIPS-published bridge between this section's deferral and the abstention of §7. RouteNLP (Guo et al., 2026a) closes a different loop at enterprise scale: conformal risk control sets per-tier escalation thresholds over token-level uncertainty (the signal of Gupta et al., 2024), and escalation failures are periodically distilled back into the cheaper tiers, converging in three iterations to quality 0.971 at 0.159 normalized cost against RouteLLM's 0.969 at 0.246 and FrugalGPT's 0.967 at 0.284, with 7.5× fewer latency-SLA violations; its own shift experiment, however, shows the marginal guarantee overshooting its 5% target to 8.1% under domain shift—the cascade-side twin of the OOD fragility documented for routers below.

Narasimhan et al. (2025) connect cascading to speculative decoding and, in doing so, make the cost of deferral precise. Their speculative cascades implement token-level deferral rules through speculative execution, and the analysis shows that the effective price of deferring at a token is proportional to the total-variation disagreement between the two models' distributions: disagreement itself is the cost. Empirically, the resulting rules dominate both lossy speculative decoding and sequential token cascades—matching large-model quality at 1.95–2.61× speedup versus 1.61–2.17×, and reaching 22.50 BLEU on WMT where quality-matched speculative decoding reaches 17.26 at equal latency. Chen et al. (2025b) push this logic to its limit and drop verification entirely: R-Stitch hands generation back and forth between a small and large reasoning model on a per-token entropy threshold, with no rollback. Their motivating measurement is a direct challenge to the verification paradigm—on low-consistency reasoning pairs, vLLM speculative decoding runs at 0.43–0.87×, an outright slowdown, because every disagreement triggers rejection, whereas entropy-routed stitching reaches 3.0–4.1× peak speedups at equal or better accuracy (up to 7.73× versus Eagle-3's $\leq$2.54×

on Qwen3-14B Minerva). Accuracy can even improve because the small model's concise traces avoid budget truncation. The two papers do not contradict so much as delimit each other: speculative cascades are evaluated on translation and summarization with matched-family drafters, R-Stitch on long chain-of-thought math with a deliberately *in*consistent small model, and which regime a deployment resembles determines whether verification is quality control or pure overhead.

When does this family work? Answer-conditioned deferral presupposes that the small model's signal separates its own errors, and the evidence above shows that separation can be bought—by tuning (Rabanser et al., 2025), calibration (Zellinger & Thomson, 2025), or sampled self-verification (Aggarwal et al., 2024)—but rarely comes for free from raw likelihoods (Gupta et al., 2024). The recurring weak point is threshold portability: every method tunes thresholds on validation data per task and budget, only Zellinger & Thomson (2025) treat that selection as a first-class statistical problem, and only Chen et al. (2025b) report a routing policy transferring across domains without retuning (their math-trained router carries to GPQA and MMLU-Redux). Two matchups the literature has not run—tuned confidences (Rabanser et al., 2025) under principled thresholds (Zellinger & Thomson, 2025), and self-verification versus confidence tuning head-to-head—are, in the design-space reading of §2.3, open combinations rather than settled questions.

**Pre-call routing: confidence as model suitability.** In pre-call routing, the system chooses a model before any candidate answer is observed. The confidence signal is therefore not answer correctness in the usual sense, but an estimate of expected model suitability for the query under a target budget or quality level. Liu et al. (2024b) make this perspective explicit by casting routing as multi-objective assignment under uncertainty: a bootstrap-ensemble classifier predicts the per-query success probability of each of 8–12 candidate LLMs, and a Pareto search over batch assignments yields 2.4–49.2% cost savings at accuracy matching the best individual model, from as little as 1% of the data as supervision. The approach presumes checkable answers to train the success predictor, which is why its evaluations stay on classification-style tasks.

RouteLLM trains routers from human preference data to choose between stronger and weaker models, and its results cut both ways in an instructive fashion (Ong et al., 2025). With judge-augmented training data, its matrix-factorization router needs only 13.4% of queries sent to GPT-4 to reach half the quality gap on MT-Bench (random routing needs 49%), translating to 3.66× cost reduction at 95% of GPT-4 quality. But the same paper shows that routers trained on Chatbot Arena preferences *alone* are approximately random on MMLU and below random on GSM8K (BERT router −11.8% APGR), and that roughly 1,500 golden-label samples—under 2% of the training data—flip MMLU to +20.7%. Supervision provenance, not router architecture, is the load-bearing choice. RouteLLM's compensating strength is transfer: because it predicts *relative* win probability rather than per-model scores, the same router carries to unseen model pairs (Claude-3, Llama-3.1) without retraining at APGR 0.767–0.772, something no pool-specific suitability model in this section can do. Router overhead is negligible throughout—$3–39 per million requests, at most 0.4% of generation cost—so the real cost of any multi-call scheme is extra generations, not the router.

Ding et al. (2024) study a more focused two-model setting in which the router predicts the quality *gap* between a small and a large model against a desired quality target. Modeling the stochasticity of generation in the labels consistently helps (their probabilistic labels dominate deterministic ones), yielding 22% fewer GPT-3.5 calls at under 1% quality drop, and at small model gaps the router can even beat all-at-large routing because the small model genuinely wins some queries. Their transfer experiment quantifies a boundary the field otherwise states loosely: a router trained on one model pair transfers when the two pairs' quality gaps correlate above 0.7 and fails outright at correlation 0.06. Barrak et al. (2025) extend routing to a pool of frontier models with no human labels at all, regressing on LLM-judge preferences and using the gap between the top-two predicted scores as an operational confidence: when the gap is small, both candidates are queried and a classifier picks the winner. That second stage is where the value is—correct-selection rises from 58.4% to 76.0% at about 0.95 extra calls per query, plateauing shortly after—and CARGO is alone in this set in auditing its own supervision, reporting inter-judge agreement of $\kappa=0.62$ and self-preference rates up to 31.1% against a 25% random baseline. Since it reports single operating points rather than cost–quality curves, its numbers are not directly comparable to the CPT/APGR literature, a small illustration of the decision-aware evaluation gap of §9.

| Method | Source | Signal | Unit | Role | Access |
|---|---|---|---|---|---|
| ● Language Model Cascades (Gupta et al., 2024) | self | token-uncertainty quantiles | answer | defer | Post |
| ○ FrugalGPT (Chen et al., 2024c) | ext | predicted utility | query / answer | route, defer | Pre, Post, Aux |
| ● AutoMix (Aggarwal et al., 2024) | hybrid | self-verification entailment | answer | defer | Post, Aux |
| ● Gatekeeper (Rabanser et al., 2025) | self | tuned small-model confidence | answer | defer | Post, FT |
| ● Rational Tuning (Zellinger & Thomson, 2025) | self | calibrated confidence + copula | stage | stop | Post |
| ● Human–AI Cascade (Fanconi & van der Schaar, 2025) | self | Bayesian-Platt $P$(True): mean + spread | answer / tier | defer, abstain to human | Post, FT |
| ● RouteNLP (Guo et al., 2026a) | hybrid | token uncertainty + conformal thresholds | query / answer / tier | route, escalate, distill | Pre, Post, Aux |
| ○ OptLLM (Liu et al., 2024b) | ext | bootstrap success-probability uncertainty | query | route | Pre, Aux |
| ○ RouteLLM (Ong et al., 2025) | ext | preference-derived relative-win probability | query / model pair | route | Pre, Aux |
| ○ Hybrid LLM (Ding et al., 2024) | ext | predicted quality gap | query | route | Pre, Aux |
| ○ CARGO (Barrak et al., 2025) | judge/ext | score gap + tie-break classifier | query | route | Pre, Aux |
| ● Self-REF (Chuang et al., 2025a) | self | confidence-token probability | answer | route, reject | Post, FT |
| ● Confidence-Driven Router (Zhang et al., 2025c) | hybrid | learned win probability from semantic-entropy labels | query / model pair | route | Pre, Aux |
| ○ Unified Routing & Cascading (Dekoninck et al., 2025) | hybrid | ex-ante + post-hoc quality | query / answer | route, defer | Pre, Post |
| ● Select-then-Route (Shah & Shridhar, 2025) | hybrid | taxonomy + judge agreement | query / answer | shortlist, defer | Pre, Post, Aux |
| ● Speculative Cascading (Narasimhan et al., 2025) | hybrid | cascade deferral score | token / stage | hand-off, stop | Mid |

Table 3: Confidence-guided model-selection methods, organized by signal source, signal form, decision unit, functional role, and operational access. Methods are marked core (●) or adjacent (○) according to the inclusion criteria C1–C3 of §2.2.

*Access:* Pre = pre-call routing; Post = post-hoc deferral after a provisional answer; Mid = mid-generation hand-off; MS = multiple sampled outputs; Aux = auxiliary router, judge, or classifier; FT = extra post-training or confidence tuning. The Confidence-Driven Router uses multiple samples only to construct semantic-entropy labels offline; its deployed router acts pre-call. The table covers the routing and cascade methods surveyed in this section, marked core or adjacent. Broader orchestration patterns such as MoA and token-stitching methods such as R-Stitch (Chen et al., 2025b) are discussed in the text but omitted here.

A different route to confidence-guided routing is developed by Zhang et al. (2025c), who replace preference labels with the models' own uncertainty: semantic entropy over sampled generations, computed offline, labels which model "wins" each training query, and those labels train otherwise standard router architectures. Like OptLLM's success-probability labels—and unlike the quality-gap and preference signals of Hybrid LLM and CARGO—the label here estimates answer reliability, which is why Table 3 marks both core while the latter remain adjacent. In the only true head-to-head of this family, at a matched training budget of 12,247 samples, their semantic-entropy-trained router reaches MT-Bench CPT(50%) of 27.3 where RouteLLM's own architectures trained on Arena preferences sit at 55.6–56.1—statistically indistinguishable from random (51.3)—and the advantage holds across all four router architectures tested, again locating the value in the label source rather than the model. Two caveats temper the result: the comparison uses RouteLLM's weakest, unaugmented configuration, and semantic entropy measures consistency rather than correctness, so the labels inherit the confidently-wrong failure mode of §7. Chuang et al. (2025a) occupy a useful middle position between routing and confidence learning: Self-REF fine-tunes the local model to emit confidence tokens after its answer, with gradient masking so the model learns to flag errors without learning the errors themselves. The resulting score routes well—matching Llama3-70B accuracy while answering locally on 61% of MMLU queries, a 2.03× latency reduction—and yields the cleanest dissociation in this section between calibration and routing utility: on Mistral, Self-REF's ECE of 0.369 routes *better* than a baseline with ECE 0.173. Better-calibrated confidence does not imply better routing, the model-selection analogue of the finding in §4 that calibration metrics misrank signals for candidate selection.

The shared failure mode of pre-call routing is distributional. Routers fail out of distribution unless their supervision covers the deployment distribution (RouteLLM's Arena-only collapse; Hybrid LLM's transfer breakdown at low gap correlation; Self-REF's domain-level pathologies, overconfident on structured technical domains and underconfident on context-heavy ones), and every pool-specific suitability model must be re-benchmarked when the pool changes—relative-win formulations are, so far, one demonstrated exception (Ong et al., 2025). A large controlled re-evaluation turns these caveats into a field-wide reality check: on LLMRouterBench's 391,645 instances, several published and commercial routers fall *below* the trivial best-single-model baseline on performance gain (FrugalGPT −9.5%, Hybrid LLM −12.7%, a commercial router −24.7%), RouteLLM barely clears it (+2.6% at 11.4% cost saving), the top performance-oriented routers are statistically indistinguishable from one another, and a twenty-point oracle gap remains, driven by model-recall failure on precisely the hard queries where routing should pay (Li et al., 2026b). Read together with RouteNLP's positive enterprise result above, the reconciliation is economic rather than algorithmic: calibrated escalation still pays where tier cost ratios are large and tasks are specialized, while generic learned routing over broad pools barely beats picking the best single model—so routing gains should be quoted against the best single model, not against random assignment.

**Hybrid systems and adjacent orchestration methods.** Several recent systems explicitly combine ex-ante routing with post-hoc deferral. Dekoninck et al. (2025) formalize the relationship and supply the timing axis of this section with its clearest evidence. Casting both mechanisms as one constrained optimization, they derive optimal strategies and a strict generalization—cascade routing, which re-routes among all remaining options after every call using updated quality estimates. On RouterBench with eleven models, cascade routing attains error–cost AUC of 87.2 versus 83.3 for pure routing and 84.5 for a FrugalGPT-style threshold cascade; on SWE-Bench the ordering is dramatic (54.1 versus 40.5 and 38.5) because binary quality feedback makes fixed thresholds degenerate; yet on their math-and-code setting plain routing beats the baseline cascade by ten points (47.5 versus 37.7). Neither paradigm dominates, and their noise analysis explains when each fails: high ex-ante estimate noise hurts routing (cascade routing gains up to +12% AUC), high post-hoc noise hurts cascading (+8%). This is the timing distinction of this section made mechanistic—route when you can predict quality before calling, cascade when you can only judge answers after seeing them—and it also derives, rather than assumes, the conditions under which simple threshold cascades are optimal.

Shah & Shridhar (2025) provide a practical realization of the hybrid view at industrial scale. SELECT-THEN-ROUTE first narrows a 24-model pool through a taxonomy classifier, then applies a cost-ordered cascade whose acceptance score fuses logit confidence, a reward model, rule-based verification, and an LLM judge. The system reports 94.3% end-to-end accuracy at $5.21 per thousand prompts, against 91.7% for the best single model (O3-Mini) and $16.29 for the pool average, and its depth ablation matches CARGO's from the routing side: the first escalation buys most of the gain (80.3%→87.1%), the second much less (89.2%), and the paper recommends stopping there. Its taxonomy is bimodal in a way that bounds the approach—near-perfect on structured tasks (98.2% on classification) but visibly weaker on open reasoning (82.0% on causal reasoning)—and, unlike CARGO, it embeds an LLM judge in the acceptance score without auditing it.

Finally, some multi-model systems are adjacent rather than central to confidence-guided routing. Wang et al. (2025a) study collaborative aggregation across models in a layered mixture-of-agents architecture, and their results are the strongest counterpoint to the selection premise of this section: an aggregator that synthesizes all proposers' answers beats an LLM ranker that merely picks the best one, and an all-open-source MoA reaches 65.1% on AlpacaEval 2.0 against GPT-4o's 57.5%. The rejoinder is cost—MoA pays $n \times l$ model calls per query, defining the spend-more-for-quality pole against which routing's spend-less-at-matched-quality framing sits. We therefore treat it as a neighboring orchestration pattern rather than a core routing or cascade method.

**Discussion.** Confidence-guided model selection extends the same basic pattern seen in earlier sections: a reliability-relevant signal is estimated for a decision unit and then converted into an action. What changes in this section is the unit itself. Instead of deciding which token, step, or answer to keep, the system now decides which model to call, whether to defer to a stronger one, or whether to continue a multi-stage pipeline. This shift makes confidence less about a single model's internal certainty and more about comparative suitability under resource constraints.

The timing axis introduced at the start of this section is not merely descriptive; it is now mechanistically supported. Pre-call routers (OptLLM, RouteLLM, Hybrid LLM, CARGO, and the Confidence-Driven Router) rely on predicted quality or score margins because no answer exists yet; for the last of these, semantic entropy supplies offline supervision rather than a runtime post-hoc signal. Post-hoc deferral methods (Language Model Cascades, AutoMix, Gatekeeper, and Self-REF) can instead condition on an actual candidate. Dekoninck et al. (2025) show that the relative noise of these two estimate types determines which paradigm wins, with double-digit AUC swings in either direction on real benchmarks. Many apparent disagreements between papers dissolve under this lens: a router that looks weak on GSM8K is often a supervision-provenance failure rather than an architectural one (Ong et al., 2025; Zhang et al., 2025c), and a cascade that underperforms routing is typically operating where post-hoc quality estimates are noisier than ex-ante ones.

Two evaluative lessons recur across the family. First, in the reported comparisons supervision provenance matters more than router architecture: judge- or gold-augmented labels rescue routers that preference data alone leaves at random (Ong et al., 2025), semantic-entropy labels beat preference labels at matched budgets (Zhang et al., 2025c), task-general self-verification transfers where dataset-specific scorers collapse (Aggarwal

et al., 2024), and confidence tuning manufactures a deferral signal that raw softmax does not provide (Rabanser et al., 2025). Second, calibration and control utility dissociate here just as they do at inference time: Self-REF routes better with worse ECE (Chuang et al., 2025a), and the overconfidence census of Zellinger & Thomson (2025) shows that the models most in need of calibration are precisely the instruction-tuned APIs most often placed at the front of cascades. Both lessons feed the decision-aware evaluation argument of §9: deferral-curve area and cost-at-matched-quality—quoted against best-single-model rather than random baselines (Li et al., 2026b)—are the metrics on which these systems actually differ, not pointwise calibration.

Finally, the strongest recent systems increasingly blur the boundary between routing and cascading. Unified formulations and shortlist-then-escalate pipelines both indicate that practical systems layer ex-ante pruning, post-hoc checking, and selective escalation—and the depth ablations agree from both sides that the first escalation step carries most of the value (Barrak et al., 2025; Shah & Shridhar, 2025). This is precisely where the broader definition of confidence adopted in this survey is useful: the controlling signal may be token uncertainty, semantic entropy, judge agreement, predicted quality gap, or bootstrap uncertainty, but in each case it is being used to decide how the system allocates model capacity.

# 6  Confidence-Gated RAG Systems

RAG introduces confidence utilization at several distinct stages of the same pipeline. In the notation of Section 2, the decision state now contains the query, any retrieved documents or snippets, partial generations, and, in some systems, verifier outputs or retrieval-quality estimates. The decision units therefore range from the query itself to documents, snippets, claims, and final answers. The corresponding actions are equally varied: decide whether retrieval is needed at all, choose a retrieval mode, filter or rerank context, verify groundedness after generation, or abstain when the resulting answer is not trustworthy enough.

What makes RAG distinctive is that confidence must be interpreted relative to *source*. A high-confidence answer may reflect strong parametric knowledge, strong support from retrieved evidence, or an unresolved conflict between the two. This is why the RAG literature repeatedly distinguishes internal knowledge from external support, and why diagnostic work such as Soudani et al. (2025) is so important: uncertainty methods that work in no-retrieval settings do not automatically transfer once retrieved context changes the model's epistemic state. The section is organized accordingly, moving from retrieval gating to context selection, then to post-hoc groundedness assessment and abstention. Table 4 summarizes where confidence enters the RAG pipeline and what decision it governs at each stage.

## 6.1  When to Retrieve

The first RAG control question is whether external evidence is needed at all. Here the relevant confidence signal is usually a query-level or generation-level estimate of the marginal value of retrieval, not a final answer confidence in the ordinary sense. Some methods make this decision during generation. Jiang et al. (2023) trigger retrieval in FLARE when predicted upcoming content contains low-confidence tokens, using those predictions to form forward-looking retrieval queries. Su et al. (2024b) broaden this idea by modeling the model's evolving information need rather than only reacting to a local token heuristic, and they explicitly improve both retrieval timing and retrieval query construction. In both cases, retrieval is activated when the ongoing generation suggests that parametric knowledge alone is no longer sufficient. The head-to-head evidence between these two gates is instructive precisely because it does not point in one direction. Re-implemented under identical retrievers and open-weight backbones, DRAGIN dominates FLARE on three of four datasets, most starkly on Vicuna-13B where FLARE's raw token-probability trigger collapses (HotpotQA EM 0.288 versus 0.092); yet FLARE remains by far the cheapest policy, averaging 1.59 retrieval calls per 2WikiMultihopQA query against DRAGIN's 2.63 (LLaMA2-13B), so DRAGIN buys its accuracy at roughly 1.6 times the retrieval budget. FLARE's original results add a caution about transfer and about retrieval itself: on text-davinci-003 it led all baselines (2WikiMultihopQA EM 51.0 versus 39.4 for single-time retrieval), and naive single-time retrieval actively *hurt* StrategyQA (EM 68.6 versus 72.9 with no retrieval at all)—but that strength does not carry over to 7–13B open models, and FLARE's own retrieval-rate sweep finds StrategyQA performance degrading once more than about half of the sentences trigger retrieval.

Other systems decide earlier, at the query level. Jeong et al. (2024) frame retrieval as strategy selection: the model should choose among no retrieval, single-step retrieval, and more elaborate iterative pipelines according to estimated question complexity. Wang et al. (2023b) use self-knowledge signals to decide whether a question lies within the model's parametric competence, so retrieval is triggered only when the model judges that outside support is needed. This is the answer-free side of the timing split of Section 5: SKR, Adaptive-RAG, and SUGAR predict retrieval need from the question before generating anything, whereas FLARE, DRAGIN, and SEAKR draft first and check the draft. The trained pre-generation gates win on single-hop efficiency rather than raw accuracy. Adaptive-RAG matches always-iterative retrieval on GPT-3.5 (EM 37.97 versus 38.13, averaged over six datasets) at 1.03 versus 2.81 retrieval steps—about 44% of the wall-clock time—and it does so with a complexity classifier that is only 54.52% accurate; an oracle classifier would add roughly 8 EM points (measured with FLAN-T5-XL), locating the remaining headroom squarely in the gate rather than the reader. SKR shows the same asymmetry at the binary level: its nearest-neighbor gate averages 68.15 accuracy versus 66.36 for always-retrieve chain-of-thought on InstructGPT, with the gains concentrated where retrieval hurts (on ChatGPT StrategyQA, always-retrieve drops accuracy from 61.36 to 57.21, and SKR recovers to 62.01). SKR's internal comparison also delivers a finding that recurs throughout this section: verbalized self-assessment is the weakest elicitation strategy, with its prompted "do you need more information?" variant falling below the nonparametric kNN gate on average. Mallen et al. (2023) provide the broader motivation for this line of work by clarifying the trust boundary between parametric and non-parametric memory, even though their paper is more analytical than algorithmic. Two recent results reframe what a pre-generation gate can be. Chen et al. (2026) show that a training-free *internal-state* signal— $P(\text{YES})$ to a can-you-answer-this probe, aggregated over the layer-by-token grid—gates retrieval, cascading, and abstention before any token is decoded, at 0.3 seconds per query: 30–600× faster than sampling-based signals at comparable AUROC, and matching always-retrieve accuracy on TriviaQA (64.0 versus 64.1) at markedly lower cost. And Marina et al. (2025) supply the counterpoint the gating literature needs: 27 LLM-free features of the question itself (entity popularity, graph statistics, precomputed knowledgability) match hybrid uncertainty gating on LLaMA-3.1-8B (mean in-accuracy 39.3 versus 39.3) at 10–20× less compute, beat it on Qwen2.5-7B, and combine with uncertainty features substitutively rather than complementarily— so much of the gating value attributed to model self-knowledge is recoverable from query-side difficulty proxies, and uncertainty-based gates must now clear this cheap external baseline.

Several recent methods make the control signal richer than a one-bit trigger. Asai et al. (2024) train SELF-RAG with reflection tokens that jointly govern whether to retrieve, whether retrieved passages are relevant and supportive, and whether a generated answer is useful. Yao et al. (2025) likewise use internal-state uncertainty as a multi-role signal that supports retrieval triggering, snippet reranking, and strategy choice. On multi-hop QA, where SEAKR re-runs FLARE, DRAGIN, and SELF-RAG under an identical retriever and corpus (SELF-RAG through its released checkpoint), the comparison is unusually clean and internal-state uncertainty wins decisively: 2WikiMultihopQA EM 30.2 versus 22.4 for DRAGIN, 14.3 for FLARE, and 4.6 for SELF-RAG. SEAKR's estimator ablation orders the signals within a fixed pipeline— Gram-determinant consistency 31.4 EM, length-normalized entropy 30.0, perplexity 29.0, verbalized self-report 27.0, energy 26.8—placing every output-level estimator below the internal-state one, with verbalized confidence near the bottom, in agreement with SKR. Reflection tokens fail on multi-hop for a distributional rather than an architectural reason: Jeong et al. (2024) independently measure SELF-RAG at MuSiQue EM 1.60 (HotpotQA 6.8, 2WikiMultihopQA 4.6), consistent with SEAKR's diagnosis that single-passage training data never taught the retrieve token to compose hops; yet SELF-RAG beats SEAKR on NQ (32.3 versus 25.6 EM), so learned gating still wins in-distribution. Li et al. (2026a) study temporal entropy patterns as another way to anticipate retrieval needs before errors fully materialize. Zubkova et al. (2025) extend adaptive gating to retrieval depth, using semantic uncertainty to choose not only whether to retrieve but also whether single-step or multi-step retrieval is warranted. Finally, Chen et al. (2025a) use agreement between two parametric answer paths as a verification signal: if the two paths converge, retrieval can be skipped; if they diverge, the system retrieves and then filters documents adaptively. Taken together, these papers show that "when to retrieve" is best understood as a decision about the expected value of non-parametric knowledge under the current state, not merely as thresholding a generic uncertainty score. Two training-side results sharpen this picture. Guo et al. (2026b) train the gate into the model with GRPO—a Brier-style reward on end-of-reasoning verbalized confidence plus an in-stream `<uncertain>` marker that

| Method | Source | Signal | Unit | Role | Access |
|---|---|---|---|---|---|
| **When to retrieve** | | | | | |
| ●FLARE (Jiang et al., 2023) | self | low-confidence future tokens | sentence / token | trigger, regenerate | Mid |
| ●DRAGIN (Su et al., 2024b) | self | information-need uncertainty | context / step | trigger, query | Mid |
| ○Adaptive-RAG (Jeong et al., 2024) | ext | predicted question complexity | query | route strategy | Pre, FT |
| ●SKR (Wang et al., 2023b) | self | self-knowledge elicitation | query | retrieve-or-skip | Pre |
| ●SELF-RAG (Asai et al., 2024) | self | reflection tokens | query / passage / answer | trigger, critique | Pre, Ctx, FT |
| ●SEAKR (Yao et al., 2025) | mech | self-aware internal uncertainty | query / snippet / strategy | trigger, rerank, route | Pre, Ctx, WB |
| ●SUGAR (Zubkova et al., 2025) | self | semantic uncertainty | query | trigger, choose depth | Pre, MS |
| ●PAIRS (Chen et al., 2025a) | self | parametric-path convergence | query / document | trigger, filter | Pre, Ctx |
| ●Internal Confidence (Chen et al., 2026) | mech | layer×token $P$(YES) probe | query | gate, route, abstain | Pre, WB |
| ○LLM-Independent AR (Marina et al., 2025) | ext | query-side popularity / graph features | query | retrieve-or-skip | Pre, Aux |
| ●Explicit-Uncertainty RL (Guo et al., 2026b) | self | trained verbal confidence + `<uncertain>` marker | answer / step | trigger, abstain | Mid, FT |
| **What Context to Keep** | | | | | |
| ●CRAG (Yan et al., 2024) | ext | retrieval-quality score | retrieval set | correct, fallback | Ctx, Aux |
| ○FILCO (Wang et al., 2023c) | ext | context usefulness score | passage / sentence | filter | Ctx, FT |
| ○InfoGain-RAG (Wang et al., 2025b) | self | document information gain | document | rerank, filter | Ctx |
| ●SKILL-RAG (Isoda, 2025) | self | self-knowledge sentence score | sentence | filter | Ctx, FT |
| ○Sparse-RAG (Zhu et al., 2025) | self | sparse document-selection score | document | select, sparsify | Ctx, Mid |
| ○UncertaintyRAG (Li et al., 2024e) | self | span uncertainty / SNR | span / chunk | score, retrieve | Ctx, FT |
| **Groundedness and Hallucination Detection** | | | | | |
| ●ReDeEP (Sun et al., 2025b) | mech | ECS + PKS | answer / mechanism | detect, mitigate | Post, WB |
| ●HALT-RAG (Goswami & Kurra, 2025) | ext | calibrated NLI ensemble | claim / answer | detect, abstain | Post, Aux |
| ●FRANQ (Fadeeva et al., 2026) | hybrid | faithfulness-conditioned factuality UQ | claim | detect | Post, Aux |
| **Abstention and Conformal Guarantees** | | | | | |
| ●TRAQ (Li et al., 2024d) | hybrid | conformal passage + answer scores | passage set / answer set | set-predict | Ctx, Post, CP |
| ●CONFLARE (Rouzrokh et al., 2024) | ext | conformal similarity threshold | chunk / retrieval set | calibrate retrieval | Ctx, CP |
| ●Principled Ctx Eng (Chakraborty et al., 2026) | ext | conformal snippet relevance | snippet | filter | Ctx, CP |
| ●Conformal-RAG (Feng et al., 2025) | hybrid | conditional conformal factuality | sub-claim | filter | Post, CP |
| ●Divide-Then-Align (Sun et al., 2025a) | hybrid | knowledge-boundary quadrant | query | abstain, align | Post, FT |

Table 4: Confidence-guided RAG methods, organized by signal source, signal form, decision unit, functional role, and operational access. Markers apply the inclusion criteria C1–C3 of §2.2: ● marks core methods whose signal is a monotone estimate of the correctness, groundedness, or reliability of its decision unit, while ○ marks adjacent methods whose estimand is difficulty, complexity, utility, or generic quality.
*Access:* Pre = before retrieval; Mid = during generation or active retrieval; Ctx = retrieved-context scoring or filtering stage; Post = after answer generation; MS = multiple samples; WB = white-box activations or logits; Aux = auxiliary evaluator, verifier, or classifier; FT = extra post-training or learned controller; CP = conformal calibration set / formal coverage guarantee. Background, dataset, and diagnostic papers such as Mallen et al. (2023); Niu et al. (2024); Lymperopoulos & Sarathy (2025); Soudani et al. (2025); Tao et al. (2024) are discussed in the text but omitted here.

triggers retrieval mid-reasoning—and beat the prompting-era gates outright on their benchmark suite: 41.6 EM at a 48.1% trigger rate against FLARE's 20.8 EM at 99.0%, DRAGIN's 32.7 at 76.8%, and SELF-RAG's 6.5, with the gains coming from better-placed rather than more retrieval. Xie et al. (2026) expose why untrained gates disappoint operationally: an uncalibrated confidence gate is not merely miscalibrated but *uncontrollable*—their base model's retrieval rate stays pinned at 25–30% across the entire threshold range—whereas after calibration-preserving SFT the same gate spans 1.6–83.8% with the threshold and captures 95% of the always-retrieve gain at 57.6% of the retrieval cost. These rankings come with sharp applicability boundaries, however: the strongest signals are the least deployable, since DRAGIN needs self-attention scores and SEAKR needs middle-layer hidden states plus twenty sampled generations per decision, both demonstrated only on open models of at most 13B and 8B parameters, respectively. Nor is adaptive triggering uniformly better than simpler policies—DRAGIN loses to single-round retrieval on IIRC (EM 0.185 versus 0.196) and to no retrieval at all on StrategyQA at 7B—so the case for confidence-gated retrieval must be conditioned on hop structure and model family rather than asserted in general.

## 6.2 What Context to Keep

Once retrieval has been invoked, the next control problem is context allocation: which retrieved units are actually worth preserving for generation? This stage is different from retrieval gating because the relevant signal now concerns retrieval quality, passage utility, or snippet coherence rather than parametric self-knowledge. Yan et al. (2024) exemplify this shift. CRAG uses a retrieval evaluator to score the quality of the retrieved set and then branches to different corrective actions: keep and refine the context when retrieval is good, fall back to web search when it is poor, or mix strategies when confidence is ambiguous. The signal therefore controls *how* to repair retrieval, not only whether retrieval should have happened. The empirical case for this externalized judgment is strong on three counts. Plugged into SELF-RAG, CRAG raises PopQA accuracy from 54.9 to 61.8 and biography FactScore from 81.2 to 86.2, and it keeps working on a plain

LLaMA2-7B generator where SELF-RAG itself collapses without its special-token fine-tuning (PopQA 29.0). Its small fine-tuned T5 evaluator reaches 84.3% judgment accuracy where prompted ChatGPT manages 58.0–64.7, a decisive win for trained lightweight judges over prompted LLM judges. And its ablations locate the value in refinement rather than triggering: removing document-level refinement costs 9.6 PopQA points (13.1 on plain LLaMA2), whereas removing any one corrective action costs at most 1.5.

Other papers make passage utility itself the scored object. Wang et al. (2025b) define Document Information Gain as the change in generation confidence caused by a document and use that signal for reranking and filtering. This is one of the clearest document-level confidence-utilization ideas in the RAG literature because the score is explicitly answer-conditioned rather than based only on semantic similarity (Table 4 nonetheless marks it adjacent: the estimand is document utility, per the boundary rule of §2.2). It is also the best-attested filter in this group: the trained reranker beats both SELF-RAG and CRAG on their shared benchmarks (LLaMA2-13B TriviaQA 76.2 versus 69.3 and 74.5; NaturalQA 51.9 versus 49.5 and 48.2) while using a single LLM call, and its ablation shows that dropping low-gain documents outright matters more than reordering them (Qwen2.5-72B 76.8 versus 73.6 without filtering). SEAKR's ablation makes the matching point from the gating side: removing its uncertainty-based re-ranking costs 2.2 EM on 2WikiMultihopQA, essentially as much as removing adaptive retrieval triggering itself (2.4)—context selection by generation utility is not an accessory to the trigger but an equal partner. Wang et al. (2023c) similarly treat context filtering as a first-class learned decision point before generation, though its learned score targets context usefulness rather than the reliability of any unit, which is why Table 4 marks it adjacent. Isoda (2025) push filtering to sentence level by using elicited self-knowledge as a signal for which retrieved content should be retained, while Zhu et al. (2025) integrate document selection into decoding itself through sparse context selection. In a broader adaptive-RAG view, both SELF-RAG and SEAKR also belong here because their reflection or uncertainty signals are reused after retrieval to critique and prioritize external evidence rather than merely to trigger retrieval.

Long-context retrieval introduces another unit of control: the span or chunk. Li et al. (2024e) show that chunk boundaries can destabilize retrieval and therefore model span-level uncertainty directly to improve similarity estimation and robustness under chunking noise. This is an important reminder that RAG confidence is not only about the final answer. It can also attach to passages, snippets, or spans whose estimated usefulness or stability determines how much external context the generator actually sees. The recurring failure mode of utility-trained filters is supervision provenance: CRAG's evaluator is trained on PopQA-style relevance with per-dataset thresholds, and information-gain labels require gold answers and by construction cannot flag confidently wrong evidence—a factually incorrect document that supports the model's answer receives positive gain. None of these filters has been evaluated on multi-hop retrieval.

## 6.3 Groundedness and Hallucination Detection

Even with retrieved evidence, RAG systems can still hallucinate by ignoring context, overtrusting parametric knowledge, or mixing the two sources improperly. At this stage the central question is no longer whether to retrieve or which evidence to keep, but whether the generated content is actually grounded in the available evidence. The field has benefited from stronger evaluation resources here. Niu et al. (2024) provide a fine-grained hallucination corpus for RAG, but the paper should be understood as dataset infrastructure rather than as a direct confidence-utilization method. Its main value in this survey is that it enabled later groundedness detectors to be studied on realistic RAG errors.

Among those detectors, Sun et al. (2025b) are especially important because they make the source conflict explicit. ReDeEP uses mechanistic signals to separate parametric knowledge use from external-evidence use, showing that hallucination often arises when internal parametric mechanisms dominate despite available context. The decoupling is what earns its numbers: on RAGTruth with LLaMA2-7B, ReDeEP reaches AUC 0.7458 against 0.7290 for the strongest external-verification baseline (RAGAS) and 0.6045 for EigenScore— the same internal-consistency family SEAKR gates retrieval on—suggesting that undecoupled internal signals conflate exactly the two knowledge sources at issue. Goswami & Kurra (2025) take a more black-box route and use calibrated NLI-ensemble features for post-hoc hallucination detection with abstention. Fadeeva et al. (2026) sharpen the target further by distinguishing factuality from faithfulness: a statement can be unsupported by the retrieved context without being false in the world, and a good uncertainty estimate

should not collapse those cases together. Conditioning uncertainty on faithfulness pays off mainly on short-form QA, where FRANQ's calibrated estimator edges every unsupervised baseline and a supervised feature ensemble (PR-AUC 0.631 versus 0.629 and 0.594 on Llama-3B); on long-form claims the absolute scores remain low (PR-AUC 0.06–0.17), which is less a criticism of the method than a measure of how unsolved claim-level factuality estimation is. This distinction is one of the most important conceptual corrections in the RAG literature.

Several adjacent papers broaden the notion of source-aware confidence. Chen et al. (2025a) compare parametric and retrieval-informed answer paths, using their convergence or divergence as a signal for whether external evidence has changed the system's belief meaningfully. Lymperopoulos & Sarathy (2025) extend uncertainty estimation to tool-augmented systems more generally, including RAG as one application, by jointly modeling LLM and tool uncertainty. Soudani et al. (2025) then provide the key diagnostic result for the whole section, one that deserves precise statement because it undercuts most of the confidence-gated pipelines above. Adding *any* document to the prompt—including a deliberately irrelevant one—significantly lowers the uncertainty of every estimator they test: predictive entropy on Llama2 falls from 1.29 with no document to 1.11 with an irrelevant document (and to 0.34 with the gold one), and the sampling-based degree score falls from 0.52 to 0.32, so these signals respond to context *presence* rather than relevance, and the axioms requiring uncertainty to rise under contradicting context are largely unmet. This raises a concrete risk for every gate that computes probability, entropy, or sample-agreement scores with retrieved context already in the prompt: FLARE's mid-generation threshold after its first retrieval, DRAGIN's information-need score on later hops, FRANQ's sampled uncertainty branches, and TRAQ's cluster-frequency nonconformity scores are all computed under exactly the conditions their axioms show deflate uncertainty—a system-level implication inferred from the estimator-level experiments rather than directly tested. It largely spares the pre-retrieval signals that classify the bare question (SKR's kNN gate, Adaptive-RAG's complexity classifier), and it leaves internal-state signals such as SEAKR's untested under in-context documents (the pre-generation probe of Chen et al. (2026) sidesteps the bias by never placing documents in the prompt)—an open empirical gap this survey flags rather than resolves. Their groundedness-aware recalibration recovers much of the damage (semantic-entropy AUROC 73.44 to 79.79 on TriviaQA), which reads, in retrospect, as an independent justification for the support tokens and retrieval evaluators that SELF-RAG and CRAG bolt onto the pipeline. Their contribution is thus not a new adaptive RAG pipeline, but a principled warning that RAG-specific uncertainty must be evaluated differently from no-retrieval uncertainty. The detectors above carry their own deployment caveats: ReDeEP requires white-box access and fits its regression weights per dataset, FRANQ needs roughly 120–300 labeled claims per calibration condition, and ReDeEP's RAGTruth evaluation presumes the retrieved context is itself accurate, so detecting failure when the context is wrong falls back to upstream evaluators like CRAG's.

## 6.4 Abstention and Conformal Guarantees

The final RAG control problem is what to do when confidence remains insufficient even after retrieval and post-hoc checking. One answer is abstention; another is to return calibrated sets or filtered claims with formal guarantees. The conformal line of work is especially important here because different papers guarantee different objects. Li et al. (2024d) apply conformal prediction to both retrieval and generation, producing passage sets and answer sets with end-to-end correctness guarantees at the set level. Rouzrokh et al. (2024) focus more narrowly on retrieval coverage by calibrating similarity thresholds so that answer-containing chunks are included with target confidence. Chakraborty et al. (2026) move the guarantee to snippet filtering before generation, while Feng et al. (2025) apply conditional conformal prediction at the sub-claim level to filter generated claims according to factuality guarantees. These papers belong together, but they should not be treated as interchangeable because they calibrate different units and guarantee different notions of success. TRAQ's numbers illustrate both the promise and the price: its end-to-end coverage holds empirically where vanilla top-1 RAG stalls at roughly 71% correctness on BioASQ regardless of the desired level, and Bayesian allocation of the error budget shrinks prediction sets by 16.2% on average over a plain Bonferroni split. But the guarantee is bought with roughly 30 sampled generations for each of up to 20 passages per question, and because its nonconformity score is a cluster-frequency signal of precisely the kind Soudani et al. (2025) show is biased by context presence, miscalibration surfaces not as broken coverage but as inflated set sizes (10.3 semantic clusters on NQ at 90% coverage).

Abstention can also be learned rather than fully conformalized. Sun et al. (2025a) model the joint knowledge boundary of parametric and retrieved knowledge and train the system to refuse when neither source is adequate. This is a confidence-utilization method in a broad sense, but its signal is knowledge-boundary membership rather than a post-hoc scalar threshold. The problem it fixes is stark: retrieval-augmented fine-tuned models never abstain at all (abstain-F1 exactly 0.00 across all three bases tested, even when prompted to refuse), although 1,241 of 3,610 NQ test queries fall outside both the parametric and the retrieved knowledge. Divide-Then-Align lifts abstain-F1 to 63.3, 59.9, and 64.7 on those bases, and its single-boundary ablations are the sharpest evidence in this section that abstention needs both source-specific signals jointly: gating on the retrieval boundary alone drops accuracy from 64.1 to 58.9, and on the parametric boundary alone to 45.8. HALT-RAG, discussed above, sits nearby as a practical precision-constrained abstention system built on calibrated verification scores. These methods show that trustworthy RAG often depends as much on deciding *not* to answer as on retrieving the right evidence. Both routes have explicit costs, however: TRAQ's guarantee rests on an i.i.d. calibration assumption and returns sets too large to act on at high coverage, while Divide-Then-Align's honesty trades away helpfulness—its denoise rate drops from 76.3 to 68.9—and exposes only a fixed refusal template, with no graded confidence for downstream composition.

More broadly, confidence-alignment work such as Tao et al. (2024) is relevant to RAG because it prepares verbalized confidence for downstream trust and retrieval decisions. However, it is best read as a bridge from confidence elicitation to RAG control rather than as a core RAG pipeline paper. That distinction matters for the survey: some methods are native RAG mechanisms, while others supply better confidence signals that RAG systems can later consume.

**Discussion.** Across these subsections, RAG makes confidence more source-sensitive than any earlier stage of the lifecycle. Before retrieval, the signal estimates whether external evidence is worth paying for. After retrieval, the signal estimates which evidence is useful enough to retain. After generation, the signal estimates whether the resulting content is grounded, faithful, or safe enough to return. The same general pattern from Section 2 still holds, but the unit and the meaning of the score change sharply from one stage to the next.

This is why source-aware distinctions are essential in RAG. Parametric self-knowledge, retrieval-quality estimates, document utility scores, mechanistic signals, NLI-based support scores, and conformal nonconformity scores are all confidence-like control signals, but they answer different questions. In particular, the literature now makes clear that faithfulness and factuality must be separated, and that confidence in internal knowledge must be distinguished from confidence grounded in retrieved context. Papers such as FRANQ, ReDeEP, PAIRS, and the axiomatic analysis of Soudani et al. (2025) are especially valuable because they expose this distinction directly rather than assuming a single scalar uncertainty can cover all cases.

The open problem is therefore not merely better calibration of one score. It is how to represent and use confidence in a way that tracks *where* the model's belief comes from. RAG has already established that confidence can gate retrieval, filter context, verify groundedness, and support abstention or formal guarantees. The next frontier is source-aware confidence that explicitly models the interaction between parametric memory and contextual evidence instead of treating them as interchangeable contributors to one undifferentiated belief state.

## 7 Confidence-Based Risk Management

Risk management is the point where confidence becomes an explicit decision variable. Given a candidate answer, a set of atomic claims, or a set-valued prediction, the system may abstain, trigger verification, flag hallucination risk, or return a calibrated set rather than a single output. In the notation of Section 2, the decision units in this section are primarily answers, claims, and prediction sets, and the downstream actions are DETECT, ABSTAIN, COVER, and CALIBRATE. The emphasis is therefore different from earlier sections: the goal is not merely to estimate confidence, but to use it to reduce failure in settings where unsupported answers are costly.

Adjacent surveys cover parts of this landscape from different angles. Rawte et al. (2023) review hallucination across foundation models, while Wen et al. (2025) survey abstention in large language models. Our focus here is narrower and more action-oriented: how confidence-like signals are obtained, and then converted into

concrete risk-control decisions. To keep that hierarchy clear, we treat confidence elicitation as an enabling layer, and then examine three main downstream families: hallucination and failure detection, conformal coverage, and abstention-oriented reliability alignment. Table 5 groups the risk-management literature by signal source, signal form, decision unit, functional role, and operational access.

## 7.1 Obtaining Actionable Confidence Signals

Before confidence can gate risk, systems need access to some actionable signal about answer reliability. Earlier sections consume confidence primarily for efficiency or accuracy, but risk management places stronger demands on signal quality, such as capturing directional overconfidence, remaining meaningful under input ambiguity, and generalizing across evaluation contexts. Foundational work introduced self-knowledge formulations such as $P(\text{True})$ and $P(\text{IK})$, separating confidence in a proposed answer from confidence that the model knows the answer at all (Kadavath et al., 2022). Closely related work evaluates self-knowledge more directly through the ability to recognize unanswerable or unknowable questions, framing uncertainty as awareness of the model's own knowledge boundary rather than only confidence in a proposed answer (Yin et al., 2023). Later work showed that, especially for RLHF models, prompted verbal or numerical confidence can be better calibrated than raw conditional probabilities, particularly when the model is asked to consider alternative answers before committing to a score (Tian et al., 2023). The apparent conflict between these two findings is narrower than it looks, because it is at least partly a post-training artifact. Kadavath et al. (2022) observe near-diagonal multiple-choice calibration only for *pretrained* models under favorable formatting—merely adding a "none of the above" option drops 52B MMLU accuracy from roughly 62% to 53% and damages calibration—and they explicitly caution that RLHF-tuned models will not be so well calibrated (in their experiments, a temperature adjustment to $T=2.5$ largely restores the RLHF policy's calibration). Tian et al. (2023) then measure exactly this RLHF regime: verbalized confidence roughly halves ECE relative to conditional probabilities (0.054 for one-stage top-4 verbalization versus 0.140 for label probabilities, GPT-3.5 on TriviaQA), and the $P(\text{True})$-style self-evaluation probability is mediocre for the same model (ECE 0.164). Two caveats from the same study bound the claim: the discrimination gains are far smaller than the calibration gains (selective AUC 0.896 versus 0.869 on TriviaQA), and the advantage is model-family-specific, with Llama-2-70B-chat showing a consistent win only on TruthfulQA. "Verbalized beats logits" is therefore best read not as a property of language models in general, but as a statement about RLHF-shifted output distributions on short-form factual QA. These signals matter in the present section not because elicitation is the end goal, but because abstention, selective generation, and post-hoc verification all depend on having some usable estimate in the first place.

However, the usefulness of an elicited signal depends on what kind of uncertainty is present. Cole et al. (2023) show that abstention is not only about epistemic ignorance; denotational ambiguity creates a separate failure mode in which the model answers confidently even though the question itself is underspecified. This is one reason why scalar calibration metrics alone are incomplete. Subsequent work sharpens this diagnosis in several directions: Net Calibration Error makes directional overconfidence explicit (Groot & Valdenegro Toro, 2024); verbal uncertainty can be traced to a representation-space feature and adjusted through mechanistic intervention (Ji et al., 2025); and consistency under uncertain-expression prompting provides a cheap black-box factuality signal without logit access (Joo et al., 2025). Even so, elicited confidence remains fragile. Verbal confidence can help in some answering settings (Tian et al., 2023), yet it remains overconfident in evaluator pipelines such as LLM-as-a-Judge (Tian et al., 2025). A recent dissociation study explains part of this fragility at the signal level: across eight models and four benchmarks, verbalized confidence predicts the model's own subsequent commit-or-abstain decision far better than it predicts correctness (median AUROC gap +0.16), and once residualized on calibrated log-probability confidence its correctness discrimination collapses to chance while its abstention prediction survives—whereas calibrated log-probabilities remain truth-anchored (Kumaran, 2026). Verbalized and distribution-derived confidence are therefore not noisy versions of one another: they carry separable variance aligned to commitment and correctness respectively, and which of the two a controller consumes determines what the control action actually optimizes. For that reason, the remainder of this section treats elicitation as a prerequisite for risk control rather than as an end in itself. As a practical rule, prompted verbalized or reformulated confidence is the elicitation route of choice when a downstream policy needs calibrated probabilities from an RLHF chat model on short-form factual questions; its documented failure modes are evaluator pipelines (Tian et al., 2025), adversarial

| Method | Source | Signal | Unit | Role | Access |
|---|---|---|---|---|---|
| **Actionable Signals** | | | | | |
| P(True) / P(IK) (Kadavath et al., 2022) | self | self-knowledge score | answer / query | elicit | Pr |
| Just Ask (Tian et al., 2023) | self | verbal / numerical confidence | answer | elicit, calibrate | Pr |
| • VUF / MUC (Ji et al., 2025) | mech | verbal-semantic mismatch | answer / hidden state | detect, steer | WB |
| • UExpr Consistency (Joo et al., 2025) | self | uncertain-expression consistency | answer | detect, flag | BB |
| **Hallucination / Failure Detection** | | | | | |
| • SelfCheckGPT (Manakul et al., 2023) | self | inter-sample consistency | sentence / answer | detect | BB, MS |
| • Semantic Entropy (Farquhar et al., 2024) | self | entropy over meaning clusters | answer set | detect | MS |
| • SEPs (Han et al., 2024) | mech | probed semantic entropy | answer / hidden state | detect | WB |
| • INSIDE (Chen et al., 2024a) | mech | EigenScore + clipping | answer / hidden state | detect | WB |
| • Internal State (Azaria & Mitchell, 2023) | mech | hidden-state truth probe | statement / answer | detect | WB |
| **Conformal Guarantees** | | | | | |
| • MCQA CP (Kumar et al., 2023) | self | option nonconformity | answer set | set-predict | CP |
| • Conformal LM (Quach et al., 2024) | self | sampling + rejection score | output set | set-predict, filter | MS, CP |
| • Conformal Factuality (Mohri & Hashimoto, 2024) | hybrid | entailment-based nonconformity | claim | filter, back off | CP |
| • Enhanced CP (Cherian et al., 2024) | hybrid | conditional factuality score | claim | filter | CP |
| • Coherent Factuality (Rubin-Toles et al., 2025) | hybrid | graph-structured conformal score | claim graph | filter, preserve coherence | CP |
| • API-Only CP (Su et al., 2024a) | hybrid | sample frequency + semantic similarity | answer set | set-predict | API, MS, CP |
| ○ ConfTS (Xi et al., 2025) | self | conformal temperature scaling | classification set | calibrate | CP |
| • CIC (Dong & Shinnou, 2026) | hybrid | any UQ score + Clopper–Pearson UCB | accepted answer | abstain, certify FDR | BB, CP |
| **Abstention / Reliability Alignment** | | | | | |
| • Self-Evaluation (Ren et al., 2023) | self | token-level self-eval | answer | abstain, select | Pr |
| • Adapt w/ Self-Eval (Chen et al., 2023a) | self | adapted self-eval score | answer | abstain | PE |
| • Selective Ambig. QA (Cole et al., 2023) | self | repetition / agreement | query / answer set | abstain | MS |
| • R-Tuning (Zhang et al., 2024b) | self | parametric knowledge boundary | query / answer | abstain, align | FT |
| • SemEnt FT (Tjandra et al., 2024) | self | semantic entropy | answer | abstain, align | FT, MS |
| • ConfTuner (Li et al., 2025d) | self | tokenized Brier verbal confidence | answer | calibrate, correct, cascade | FT |
| • FISCORE (An & Xu, 2025) | self | semantic cluster consensus | answer sample | abstain, align | RL, MS |
| • TruthRL (Wei et al., 2026) | hybrid | ternary correct/abstain/hallucinate reward | response | abstain, align | RL |
| • Act.-Based Abst. (Huang et al., 2025c) | mech | FFN activation confidence | answer | abstain | WB |

Table 5: Confidence-based risk-management methods, organized by signal source, signal form, decision unit, functional role, and operational access. Following the inclusion criteria of §2.2, • marks core methods whose confidence signal is consumed in a risk-control decision (criteria C1–C3 with correctness of the unit as the target property). Unmarked rows (the pure elicitation primitives of Kadavath et al., 2022 and Tian et al., 2023) fail C2 in isolation; they are listed as background because the decision layers above consume their signals. Detectors whose score is thresholded into a system-facing flag satisfy C2 and are marked accordingly (§2.2).

*Access:* Pr = prompted elicitation or self-evaluation; BB = black-box prompting only; WB = white-box hidden states or logits; MS = multiple samples; API = API-only deployment; PE = parameter-efficient adaptation; FT = extra fine-tuning; RL = reinforcement learning; CP = conformal guarantee. Diagnostic metrics, surveys, benchmarks, and adjacent papers such as NCE, LLM-as-a-Judge overconfidence, FACTSCORE, SafeBehavior (Zhao et al., 2025), Li et al. (2025a), Tripathi et al. (2025), and Wen et al. (2025) are omitted here.

prompting, and reasoning-heavy or long-form tasks, which Tian et al. (2023) explicitly leave untested. When the consumer instead needs discrimination—ranking likely errors—rather than calibrated probabilities, the sampling-based detectors of §7.2 are the stronger family.

## 7.2 Hallucination and Failure Detection

One major use of confidence in risk management is to identify outputs whose factuality is doubtful before they are acted upon downstream. Black-box methods exploit behavioral instability. Manakul et al. (2023) use divergence across sampled responses as a zero-resource hallucination signal, showing that self-consistency alone can separate factual from non-factual statements. Farquhar et al. (2024) replace raw lexical disagreement with semantic entropy, clustering sampled answers by meaning and measuring uncertainty over semantic clusters rather than surface forms. This shift is important because many incorrect outputs differ only lexically, whereas semantic entropy targets confabulations at the level of meaning.

Head-to-head numbers support a rough ordering of signal families, but only within a fixed evaluation protocol. Averaged over thirty model–dataset combinations of short-form QA, semantic entropy attains AUROC 0.790, against 0.698 for $P(\text{True})$, 0.691 for naive length-normalized entropy, and 0.687 for a supervised embedding-regression probe (Farquhar et al., 2024); on WikiBio passages, SelfCheckGPT's LLM-prompt variant reaches sentence-level AUC-PR 93.42 (NLI variant 92.50), far above the best grey-box probability baseline at 83.21 (Manakul et al., 2023). This ordering does not survive a change of estimator and judge, however. Under the protocol of Chen et al. (2024a) on CoQA—a different semantic-entropy estimator, sampling at $T=0.5$, and ROUGE-based rather than entailment-based correctness labels—semantic entropy scores only 63.1–65.3

AUROC while their EigenScore reaches 71.2–72.8 and SelfCheckGPT 68.1–70.2; and on TriviaQA, plain perplexity wins outright (83.6 versus 82.7 for EigenScore on LLaMA-7B) because one-or-two-word answers make token-level confidence sufficient. Detector rankings are therefore benchmark- and estimator-conditional: reported AUROCs cannot be compared across papers without matching the sampling temperature, sample count, entropy estimator, and correctness judge.

Later work compresses or redirects these signals for deployment. Han et al. (2024) approximate semantic entropy with cheap probes over hidden states from a single generation, preserving much of the original detector's strength while removing repeated sampling at test time: out of distribution (leave-one-dataset-out), these semantic-entropy probes beat correctness-supervised probes by 7.7–10.5 AUROC points on short-form QA (2.2–6.2 on long-form), yet remain slightly below the ten-sample semantic-entropy signal they distill (about 0.78 versus 0.83 on BioASQ for Llama-2-7B)—quantifying both the cost saving and the residual gap. Chen et al. (2024a) and Azaria & Mitchell (2023) instead access internal representations directly, using hidden-state classifiers or semantic-space consistency scores to detect false or overconfident generations even when the decoded text looks fluent; the cost profiles differ sharply, with EigenScore running at about 0.81 seconds per question versus 10.68 for SelfCheckGPT (roughly 13×), although that comparison is against SelfCheckGPT's weakest (BERTScore) variant (Chen et al., 2024a). At the black-box end, Joo et al. (2025) use consistency under uncertain-expression prompting as a lower-cost factuality cue, while Muhammed et al. (2026) show that many of these detectors transfer unevenly across domains, especially from biography-style factuality to mathematical reasoning. In long-form settings, frameworks such as FACTSCORE provide the atomic-claim decomposition that many later detection and conformal methods rely on, even though they are primarily evaluation infrastructure rather than detectors themselves (Min et al., 2023).

In deployment terms: with a sampling budget and either logits or an entailment judge, semantic entropy is the strongest general-purpose detector under entailment-judged short-form protocols, while single-pass probes trained on semantic-entropy labels are the best cost-adjusted choice under white-box access and degrade most gracefully off-distribution, exactly where correctness-supervised probes collapse (Farquhar et al., 2024; Han et al., 2024). The stated failure modes are shared: consistency-based detectors target *arbitrary* confabulations and by construction miss consistently wrong learned errors (Farquhar et al., 2024), degrade sharply at sampling temperatures above one (Chen et al., 2024a), and carry a roughly 5–20× sampling overhead unless probed or distilled.

## 7.3 Conformal Prediction and Coverage Guarantees

Conformal methods use confidence or nonconformity scores in a different way. Rather than ranking answers by trust alone, they convert uncertainty into set-valued outputs or filtered responses with explicit coverage guarantees. Early LLM work applies split conformal prediction to multiple-choice QA, producing answer sets whose size reflects uncertainty and whose guarantees depend on the usual exchangeability assumptions (Kumar et al., 2023). For open-ended generation, conformal language modeling calibrates sampling and rejection rules over candidate outputs (Quach et al., 2024), while API-only methods replace logits with sampling frequency and semantic similarity so that conformal prediction remains possible in black-box deployments (Su et al., 2024a).

Another line targets factuality at the claim level. Mohri & Hashimoto (2024) introduce conformal factuality, progressively backing off from a response by removing the least certain claims until the remaining output satisfies a target correctness level. Subsequent work strengthens this family in two directions: Cherian et al. (2024) make the guarantees more adaptive and topic-sensitive, and Rubin-Toles et al. (2025) extend conformal factuality to reasoning tasks where claims cannot be filtered independently but must remain coherent as a graph.

A fourth guarantee target has recently joined this family: Dong & Shinnou (2026) certify the *acceptance-conditioned error rate*—the error rate among answers the system actually returns—by choosing an abstention threshold over any black-box uncertainty score through Clopper–Pearson upper confidence bounds, so that the false-discovery rate among accepted answers stays below $\alpha$ with probability $1 - \delta$ (e.g., realized FDR 0.037 at target 0.05 on TriviaQA with Qwen2.5-7B), and the procedure honestly reports infeasibility when the base model cannot support the requested risk level.

These methods guarantee different units, and the unit determines the deployment. Multiple-choice conformal prediction (Kumar et al., 2023), like the classification setting of Xi et al. (2025), guarantees *label coverage*—the true label lies in the returned set with probability at least $1 - \alpha$—so its natural consumer is a system that can act on a shortlist of options. Conformal language modeling (Quach et al., 2024) guarantees that the returned set *contains at least one admissible response*, with a separate bound on erroneously retained components; this fits human-in-the-loop triage (its motivating radiology setting hands a reviewer a calibrated set of candidate reports), and its calibrated stopping rule doubles as budget control, with likelihood-based set scoring achieving less than half the set size of count-based stopping on TriviaQA (normalized size 0.08 versus 0.21). Conformal factuality (Mohri & Hashimoto, 2024) instead guarantees that the *single emitted output* is factual at level $1 - \alpha$, which suits autonomous single-answer emission; the consequence is content loss, since raising GPT-4 biography factuality from roughly 25–30% to a guaranteed 80% retains only about half of the sub-claims—and sampling-frequency scores prune better than GPT-4's own verbalized confidence when ranking claims for removal.

A key lesson from this literature is that conventional calibration and conformal efficiency are not the same objective, and the effect size is large. On CIFAR-100 with ResNet-50 at $\alpha$=0.1—a classification testbed, not free-form generation—every standard calibration method *lowers* ECE yet *enlarges* adaptive (APS) conformal sets, with temperature scaling growing average set size from 4.91 to 6.69; on ImageNet, temperature scaling enlarges APS sets from 9.06 to 12.1 even as ECE improves from 3.69 to 2.24, while coverage is unaffected throughout, so the damage is invisible to validity checks (Xi et al., 2025). The efficiency-targeted fix from the same work, ConfTS, retunes the temperature against a conformal efficiency gap rather than ECE, shrinking ViT-B/16 APS sets from 36.72 to 9.05 at $\alpha$=0.05 and improving average APS efficiency by 58.3% across six ImageNet models at $\alpha$=0.1. Because these experiments concern image and text classification rather than LLM generation, Table 5 marks ConfTS adjacent: it supplies cross-domain evidence for the objective mismatch, while transfer to free-form generation sets remains untested. General conformal risk control provides the theoretical backdrop, but the practical design choice in LLMs is the unit to which the guarantee is attached: answer options, sampled outputs, or atomic claims (Angelopoulos et al., 2024). Relatedly, Gui et al. (2024) treat trustworthiness itself as the conformal target, using a calibrated threshold to select outputs whose predicted alignment scores imply a user-specified reliability level. A complementary recent direction, Conformal Linguistic Calibration, reinterprets linguistically hedged answers as answer-set prediction and uses conformal guarantees to trade off factuality against specificity, thereby linking coverage control to verbal uncertainty expression more directly (Jiang et al., 2025).

Two limitations recur in this family's own statements of scope: the guarantees are marginal rather than conditional, so per-topic or per-user coverage can deviate, and they assume exchangeable prompts, excluding distribution shift and multi-turn interaction (Mohri & Hashimoto, 2024; Quach et al., 2024). Conformal wrappers are therefore the right tool when an auditable average error rate is required, with validity resting on the exchangeability assumptions just stated; their *usefulness*—claims kept, set size, samples drawn—is inherited entirely from the quality of the inner confidence score, and none of these methods has yet been run with semantic-entropy-grade inner scores, an unmeasured efficiency opportunity.

### 7.4 Abstention and Reliability Alignment

Abstention methods use confidence more directly: the model answers when the signal suggests adequate support and refuses, defers, or hedges when it does not. In selective generation, raw sequence likelihood is not merely weak but below chance as an abstention signal—calibration AUC 39.8 on TruthfulQA with PaLM-2 Large—whereas reformulating the same logits into token-level self-evaluation restores it to 73.8, or 75.3 for a hybrid variant that adds a none-of-the-above score (Ren et al., 2023); the failure lies in the decision unit (whole sequences), not in logits per se, though the margins shrink to near chance (about 55) for holistic summary quality. Parameter-efficient adaptation can further improve the quality of those self-evaluation signals for task-specific selective prediction (Chen et al., 2023a). R-Tuning turns abstention into a training objective by constructing refusal-aware supervision around the model's parametric knowledge boundary, improving average precision by roughly 17 points over vanilla instruction tuning on in-domain MMLU (68.87 versus 51.93 with LLaMA-13B); notably, in the cleanest train-time-versus-post-hoc comparison in this literature, its unsupervised variant beats applying the *identical* sampling-consistency signal as a

post-hoc test-time filter (Zhang et al., 2024b). Semantic-entropy fine-tuning then replaces R-Tuning's label-matching supervision with a label-free semantic-entropy split and Pareto-dominates both R-Tuning variants across abstention thresholds on sentence-length generation, reducing accuracy–engagement distance by up to 30.1%, while R-Tuning's lexical-entropy variant collapses on long-form outputs (0.521 versus 0.380 for the untuned model) because every sampled sentence is a distinct string (Tjandra et al., 2024). This pair of studies also shows how the evaluation metric can flip a comparison's sign: under the average-precision metric of Zhang et al. (2024b), refusal tuning clearly beats its baselines, whereas under the accuracy–engagement metric of Tjandra et al. (2024), which explicitly penalizes over-refusal, the R-Tuning variants are often worse than the untuned instruct model. Whether refusal tuning "works" is thus metric-dependent, not merely method-dependent.

The training-side frontier has since moved to reinforcement learning, and the metric question has acquired a formal backdrop. TruthRL (Wei et al., 2026) replaces refusal-aware supervision with a ternary GRPO reward (+1 correct, 0 abstain, −1 hallucinate) whose group-relative advantage credits abstention only relative to the alternatives actually sampled: on CRAG with retrieval it cuts hallucination from 43.5% to 19.4%, and where R-Tuning collapses to over-abstention out of distribution (91.3% abstention on GSM8K, truthfulness −4.0), TruthRL abstains on only 2.7% at truthfulness 71.9—evidence that static knowledge-boundary labels, not abstention training itself, cause over-conservatism. Meanwhile Kalai et al. (2025) prove that under binary grading abstention is strictly suboptimal for any belief state and document that nine of ten dominant benchmarks award zero credit to "I don't know"—so every abstention-tuned model in this subsection fights an incentive gradient that leaderboard-optimizing post-training reimposes, and their proposed fix (explicit confidence targets with penalty $t/(1-t)$, audited by behavioral calibration) is a population-level instance of the decision-aware evaluation we advocate in §9.

More recent work refines the abstention signal itself. Huang et al. (2025c) use activation-based uncertainty estimation to support low-latency response abstention in high-stakes RAG deployments, Li et al. (2025d) train models to verbalize calibrated confidence through a tokenized Brier objective, and An & Xu (2025) use fine-grained semantic consensus as a reward for abstention-oriented reinforcement learning. These methods are best understood as reliability alignment rather than pure calibration: they attempt to move the refusal boundary itself, not merely to post-process a fixed score. That distinction is central in the abstention literature surveyed by Wen et al. (2025), which emphasizes that query answerability, model confidence, and value alignment can all justify refusal. It is also why robustness matters. Verbal confidence may be useful under nominal prompting, yet it remains vulnerable to adversarial manipulation (Obadinma & Zhu, 2025), and broader work on calibration attacks suggests that confidence robustness is itself a separate design objective rather than a by-product of nominal calibration (Obadinma et al., 2024). We return to this issue in Section 9 (Challenge 5), where we discuss robustness and portability as a cross-cutting open problem. In deployment terms, train-time abstention binds a binary refusal policy into the weights—the operating point is fixed when the tuning data are built, and fine-tuning access is required—whereas post-hoc scores let the deployer move the threshold per application. The stated limitations of both training-based studies (crude proxies for parametric knowledge, prompt-format sensitivity, small LoRA budgets, single base models) caution against assuming refusal tuning transfers across model families.

**Discussion.** The preceding subsections suggest that confidence-based risk management serves at least three distinct objectives. Calibration-oriented work asks whether a signal tracks empirical correctness or uncertainty. Selective-reliability work asks whether the system abstains, filters, or escalates at the right boundary. Conformal work asks whether set-valued or filtered outputs achieve a target error rate while remaining useful. These objectives are related, but they are not interchangeable, and the literature becomes much easier to read once that distinction is made explicit.

This perspective also explains several apparent tensions in the literature. Verbalized confidence can out-perform raw probabilities in some RLHF answering settings (Tian et al., 2023) yet remain overconfident in evaluator pipelines or under adversarial prompting (Tian et al., 2025; Obadinma & Zhu, 2025). Calibration-focused fine-tuning can improve verbal honesty or ECE (Li et al., 2025d) without improving conformal efficiency (Xi et al., 2025). Likewise, methods that reshape the refusal boundary through tuning or reinforcement learning (Zhang et al., 2024b; Tjandra et al., 2024; An & Xu, 2025) solve a different problem

from post-hoc detectors such as semantic entropy or SelfCheckGPT (Farquhar et al., 2024; Manakul et al., 2023). The practical question is therefore not which method is "most calibrated" in the abstract, but which guarantee the deployment setting actually requires.

In practice, disagreement- and representation-based detectors are appropriate when the goal is to warn on likely hallucinations; abstention-alignment methods are more suitable when the system must refuse unsupported answers; and conformal methods are the right tool when auditable coverage guarantees are required, even at the cost of larger sets or less specific outputs. Across all three settings, the open problem is joint design: obtaining confidence signals that remain informative under ambiguity, robust under attack, and useful for downstream control rather than only for post-hoc reporting.

# 8 Confidence in Agentic Systems

Confidence-guided agentic control extends the survey from single decisions to composed action loops. In agent settings, the system must decide not only which answer to trust, but whether to call a tool, allocate more compute, revise a partial plan, backtrack from an error state, or aggregate multiple agents into a final action. Under the broad definition used throughout this survey, the relevant signals therefore include self-confidence, verifier uncertainty, process-reward estimates, hesitation features, and communicated peer confidence. The common pattern from Section 2 still applies: a reliability-relevant score is attached to a decision unit and then converted into an action. What changes is that the unit itself can now be a tool call, a reasoning step, a search branch, a full trajectory, or a team-level vote.

**Extensions to the core notation.** Agentic control benefits from two additional descriptors. The first is *temporal horizon*: confidence may attach to a local step, an intermediate branch, a full trajectory, or an end-to-end episode. The second is *actor scope*: the relevant signal may come from the acting agent itself, from a verifier or reward model, from peer agents in a discussion, or from an external orchestrator. In practice, the state $\xi_t$ now contains not only the prompt and partial output, but also environment observations, memory, tool outputs, and peer messages. This section uses those two descriptors to organize the literature into four layers of control: selective escalation, self-correction, verifier-guided search, and multi-agent aggregation. Table 6 collects the agentic control methods, marked core or adjacent per the criteria of §2.2, and highlights which signals drive escalation, search, or aggregation.

## 8.1 Selective Escalation, Tool Use, and Budget Allocation

The first agentic control problem is whether additional action is warranted at all. Rather than always debating, always searching, or always invoking more compute, several recent systems use confidence-like signals to decide when escalation is likely to pay off. Fan et al. (2026) study this question directly for multi-agent debate. Instead of triggering debate through a raw confidence threshold, they extract structured hesitation features from a self-critique and train a lightweight classifier to decide whether debate is likely to improve the answer. The signal is therefore query-level and policy-facing: invoke a more expensive collaborative procedure only when the expected gain justifies the cost.

Adaptive compute allocation makes the same decision at a different granularity. Park et al. (2025) calibrate process reward model (PRM) success estimates so that they can be used as decision-grade confidence signals for allocating additional search or best-of-$N$ budget. Snell et al. (2025) frame the broader systems problem: inference-time compute should be allocated unevenly across prompts, since easy queries benefit less from extra search than medium-difficulty ones. In both cases, the decision is not which answer is currently best, but whether a partial trajectory or prompt deserves more budget. Verification-aware planning systems such as Xu et al. (2026) push this logic into coordination itself, using planner-defined verification functions to decide whether a subtask should proceed, retry, or trigger replanning.

This selective-escalation view also clarifies what does *not* belong centrally in this section. Confidence-gated edge-cloud routing, dynamic retrieval, and answer rejection are closely related ideas, but papers such as Zhang et al. (2025c); Chuang et al. (2025b); Jin et al. (2025); Chuang et al. (2025a); Fu et al. (2025) are more naturally treated in Sections 5, 6, and 4, where the controlled units are models, retrieved context, or candidate answers rather than native agent actions.

| Method | Source | Signal | Unit | Role | Access |
|---|---|---|---|---|---|
| **Selective Escalation** | | | | | |
| ○ iMAD (Fan et al., 2026) | self | hesitation-feature trigger | query | trigger debate | Pr, FT, MA |
| ● PRM Calibration (Park et al., 2025) | auxiliary | calibrated PRM success | prefix / traj | allocate budget | Aux, MS, FT |
| ○ Scaling TTC (Snell et al., 2025) | hybrid | difficulty / verifier value | query / traj | allocate compute | Aux, MS |
| ● VeriMAP (Xu et al., 2026) | auxiliary | planner verification funcs | subtask / plan node | retry, replan | Env, Aux |
| **Self-Correction / Backtracking** | | | | | |
| ● ReVISE (Lee et al., 2025) | self | intrinsic self-verification | step / traj | revise, stop | FT |
| ● SSR (Shi et al., 2025) | self | step self-consistency | Socratic step | locate, refine | MS, Pr |
| ● BacktrackAgent (Wu et al., 2025c) | auxiliary | verifier + judger score | page / action state | detect, backtrack | Env, Aux |
| **Verifier- and Reward-Guided Search** | | | | | |
| ○ LATS (Zhou et al., 2024) | hybrid | value + reflection signal | node / traj | expand, select | Env, MS |
| ○ ReST-MCTS (Zhang et al., 2024a) | auxiliary | inferred process reward | step / traj | expand, filter | MS, FT |
| ● UATS (Song et al., 2026) | auxiliary | PRM uncertainty (MC Dropout) | step / node | guide search, allocate budget | Aux, MS, FT |
| ○ AgentRM (Xia et al., 2025) | auxiliary | trajectory reward score | step / traj | guide search | Aux, FT |
| ○ RewardAgent (Peng et al., 2025) | hybrid | RM + verifier reward | response / traj | select, supervise | Aux, FT |
| **Multi-Agent Deliberation** | | | | | |
| ● ReConcile (Chen et al., 2024b) | peer | agent confidence + explanations | answer / round | aggregate, update | MA |
| ● ConfMAD (Lin & Hooi, 2025) | peer | LN / verbal confidence | response / turn | aggregate, revise | MA |

Table 6: Confidence-guided agentic control methods, organized by signal source, signal form, decision unit, functional role, and operational access. Methods are marked core (●) or adjacent (○) according to the inclusion criteria C1–C3 of §2.2.

*Access:* Pr = prompted self-critique or self-evaluation; Env = interactive environment or planner-mediated execution; Aux = external verifier, reward model, or planner module; MS = multiple samples / rollouts; FT = extra post-training or learned controller; MA = multi-agent discussion. Baseline or diagnostic papers such as Madaan et al. (2023); Shinn et al. (2023); Kim et al. (2025); Yang et al. (2025); Cui et al. (2026); Kaesberg et al. (2025); Wu et al. (2025a) and off-section routing / RAG papers such as Zhang et al. (2025c); Chuang et al. (2025b); Jin et al. (2025); Chuang et al. (2025a); Fu et al. (2025) are discussed in the text but omitted here.

## 8.2 Self-Correction, Revision, and Backtracking

The next control layer operates inside an ongoing trajectory. Foundational baselines such as Madaan et al. (2023) and Shinn et al. (2023) show that language feedback can improve future behavior, either within a single critique-and-revise loop or across repeated trials with memory. These papers are essential context, but their signals are mostly holistic and verbal: the model critiques an output or stores a lesson, rather than explicitly deciding which local part of a trajectory is unreliable.

Several later papers make that decision much more explicit. Lee et al. (2025) learn intrinsic self-verification and an internal stop-versus-revise decision, then use the resulting verification confidence both to determine whether refinement should continue and to bias inference-time decoding. Shi et al. (2025) move from whole-answer critique to process-level repair by decomposing reasoning into Socratic steps, estimating confidence for each step through repeated sub-question solving, and revising from the weakest point. In interactive environments, Wu et al. (2025c) use verifier and judger signals to decide whether the current GUI state indicates an error that warrants backtracking. What these papers share is not merely self-improvement, but explicit local control: confidence or verification is attached to a step or state, and that signal determines whether the agent should continue, revise, or roll back.

Other agentic papers are better understood as *behavioral* rather than *calibration-native* control. Kim et al. (2025) teach reflection and backtracking from search-derived traces, while Yang et al. (2025) explicitly train models to emit backtracking actions during reasoning. These methods clearly govern downstream actions, but the control signal is embedded in learned search behavior rather than surfaced as a calibrated scalar. Distinguishing these cases matters for the survey: they are relevant agentic control papers, but they should not be presented as if they solve the same problem as intrinsic self-verification methods such as ReVISE or step-confidence methods such as SSR.

## 8.3 Verifier- and Reward-Guided Search

Search-based agent systems attach control signals to prefixes, nodes, or trajectories rather than final answers. Zhou et al. (2024) combine MCTS with reflections and environment feedback so that expansion decisions depend on value-like estimates over future trajectories. Zhang et al. (2024a) similarly use inferred process

rewards to guide tree search and to filter self-training traces, treating partial reasoning quality as a signal for which branches merit further exploration. These are broad control frameworks rather than narrow confidence-estimation papers, but under the survey's operational definition they belong here because the reward or value signal directly governs search actions.

The more confidence-native part of this literature asks whether the controller itself can be trusted. Song et al. (2026) quantify epistemic uncertainty in process reward models via Monte Carlo Dropout, showing that PRMs can be overconfident on out-of-distribution reasoning paths, and propose an RL-based controller that dynamically allocates search budget based on that uncertainty. Park et al. (2025) make the same issue operational by calibrating PRM success estimates before using them for adaptive inference-time scaling. Xia et al. (2025) and Peng et al. (2025) extend this logic to learned reward systems, where trajectory-level or response-level reward signals are used for best-of-$N$ selection, beam search, or preference construction. Across these papers, the recurring lesson is that once an agent's search policy depends on a verifier or reward estimate, uncertainty in the verifier becomes part of the control problem itself.

## 8.4 Multi-Agent Deliberation and Aggregation

In multi-agent systems, confidence becomes a communicated social signal rather than a purely internal estimate. Chen et al. (2024b) combine diverse models, per-agent confidence, and calibrated confidence-weighted voting, arguing that debate works best when confidence is paired with model diversity and convincing corrective explanations. Lin & Hooi (2025) make this dependence even more explicit by comparing token-probability-derived and self-verbalized confidence, then studying how calibration changes debate quality. Fan et al. (2026) move one step earlier and ask whether debate should happen at all, using self-critique-derived hesitation features to trigger multi-agent deliberation only when it is expected to help.

At the same time, the strongest diagnostic papers caution against simplistic narratives. Wu et al. (2025a) show in a controlled setting that diversity and intrinsic reasoning strength matter more than visible confidence alone, and that debate can fail through majority pressure even when confidence is exposed. Kaesberg et al. (2025) and Cui et al. (2026) further show that aggregation protocol is not an implementation detail: task type, conformity pressure, and trajectory-level scoring all change whether collaboration helps or harms. The practical lesson is that confidence in debate is only useful when the surrounding protocol knows how to interpret it. Calibrated local confidence does not automatically imply calibrated collective behavior.

**Discussion.** Agentic control extends confidence utilization from one-step decisions to layered control policies over time. The key distinctions are temporal horizon, actor scope, and signal type: some methods rely on self-verification, some on verifier uncertainty or reward models, and some on communicated peer confidence. These distinctions clarify why papers such as ReVISE, SSR, UATS, PRM Calibration, ReConcile, and ConfMAD are core examples of confidence utilization, whereas papers such as Self-Refine, Reflexion, LATS, or Free-MAD are better understood as baselines or adjacent control frameworks with weaker or more implicit confidence semantics.

The main open problem is composition, and the notation of §2 makes its structure precise. A realistic agent stacks at least three signal–policy couplings: a trigger signal $\kappa^{(1)}$ decides whether to escalate at all (Fan et al., 2026), a step- or node-level signal $\kappa^{(2)}$ guides search or revision (Song et al., 2026; Zhang et al., 2024a), and an aggregation-level signal $\kappa^{(3)}$ combines proposals into a final action (Chen et al., 2024b; Lin & Hooi, 2025). Current work calibrates each layer in isolation, but the layers are not independent: each downstream signal is evaluated only on the units that survived the upstream gate, so its effective input distribution is a selection-biased residual of the original one. A process reward model that is calibrated on unfiltered trajectories can therefore be systematically overconfident on exactly the trajectories that reach it in deployment—a compounding of the out-of-distribution overconfidence that Song et al. (2026) document for PRMs even before any upstream gating is applied. The same argument runs in the opposite direction at aggregation time: confidence-weighted voting assumes the incoming per-agent confidences are comparable, yet upstream self-correction loops (Lee et al., 2025; Shi et al., 2025) modify each agent's confidence distribution differently, which is one reading of why exposing confidence does not reliably improve debate outcomes in controlled studies (Wu et al., 2025a; Kaesberg et al., 2025). Three concrete directions follow. First, *joint* calibration of stacked gates on the post-selection distributions they actually face, rather than per-layer calibration on

unconditioned data (Park et al., 2025). Second, propagation of the controller's own uncertainty: when the verifier is uncertain about its score, that second-order signal should widen search rather than being collapsed into the point estimate (Song et al., 2026). Third, protocol-aware aggregation that treats communicated peer confidence as a signal whose semantics depend on the emitting agent's upstream pipeline, not as a common currency (Chen et al., 2024b; Cui et al., 2026). Reliable agentic systems will require confidence and verification signals that remain meaningful as they propagate across tools, steps, trajectories, and interacting agents; we return to this as Challenge 2 in §9.

## 9 Open Challenges

Section 2 framed confidence utilization as a decision process in which a reliability-relevant signal $\kappa_t(u; \xi_t)$ is attached to a unit $u$ under state $\xi_t$, optionally transformed, and then consumed by a policy that changes the system's behavior. Across training, inference, routing, retrieval, risk control, and agentic systems, this abstraction proved broad enough to cover log-probability-derived scores, uncertainty estimates, agreement statistics, verbalized confidence, verifier outputs, reward-model scores, and peer-reported confidence. At the same time, the survey makes clear that the field still lacks a mature theory of how such signals should be interpreted, compared, and deployed. As summarized in Figure 5, we highlight five open challenges that recur across the full lifecycle.

**Challenge 1: Heterogeneous confidence semantics.** The survey deliberately uses "confidence" in a broad operational sense, but that breadth exposes a foundational problem: many useful control signals do not estimate the same underlying quantity. Verbalized confidence can outperform raw conditional probabilities for RLHF-tuned models (Tian et al., 2023); agreement across sampled reasoning paths can act as a practical reliability proxy (Wang et al., 2023a); semantic entropy measures uncertainty over meanings rather than strings (Farquhar et al., 2024); calibrated process reward models estimate the future success probability of a partial trajectory (Park et al., 2025); and debate systems surface peer confidence as a communication signal rather than a private scalar (Chen et al., 2024b; Lin & Hooi, 2025). All of these signals are operationally useful, yet they are not obviously commensurate. Recent evidence upgrades this concern from a caution to a measured dissociation: verbalized confidence predicts a model's own subsequent commit-or-abstain decision far better than it predicts correctness, while calibrated log-probabilities remain truth-anchored (Kumaran, 2026)—so two "confidence" signals from the same model can track different latent quantities. A central open problem is therefore semantic: when two methods expose different confidence signals, when should those signals be interpreted as competing estimates of the same property, and when do they represent fundamentally different objects? Satisfying criterion C1 in practice requires verifying *which* quantity a controller actually consumes.

**Challenge 2: Composition across units and horizons.** Confidence utilization rarely ends at the level where the signal is first computed. Local signals must usually be lifted into broader decisions: low-confidence token groups can trigger regeneration or early stopping (Fu et al., 2025), weak reasoning steps can trigger local repair (Shi et al., 2025), calibrated prefix-success estimates can decide how much additional search a trajectory deserves (Park et al., 2025), and confidence-weighted votes can decide the outcome of multi-agent discussion (Chen et al., 2024b). Yet there is still no general principle for propagating reliability from tokens to answers, from steps to trajectories, or from individual agents to collective action. Existing systems typically rely on heuristic aggregation, repeated sampling, or pipeline-specific controllers. A major open direction is to develop principled composition rules for turning local confidence into global control policies without losing the information that made the local signal useful in the first place.

**Challenge 3: Source attribution and confidence fusion.** The strongest systems increasingly combine several confidence sources at once, but they rarely preserve the provenance of those signals. This is especially visible in retrieval-augmented generation, where ReDeEP shows that parametric knowledge and retrieved evidence can disagree internally, and that accurate diagnosis may require disentangling the two sources mechanistically (Sun et al., 2025b). Similar source-allocation problems appear elsewhere: routing systems may combine self-reported confidence with external routers (Chuang et al., 2025a); adaptive inference may rely on verifier or reward-model scores whose own uncertainty must be modeled (Park et al., 2025); and

| 1 Heterogeneous semantics | 2 Composition | 3 Source attribution | 4 Decision-aware evaluation | 5 Robustness & portability |
|---|---|---|---|---|
| agreement, entropy, verifier and verbal scores estimate different $\rho$; when are they commensurate? | lifting $\kappa$ from tokens $\rightarrow$ answers $\rightarrow$ trajectories $\rightarrow$ teams without losing information | fusing parametric, retrieved, and peer signals while preserving provenance | metrics indexed by the action $\delta$ controls, not one calibration score | do signals survive model swap, prompt shift, and adversarial pressure? |

Figure 5: Five open challenges that recur across the lifecycle, each phrased in the notation of §2: heterogeneous semantics of $\kappa$, composition of $\kappa$ across units and horizons, provenance-aware fusion of multiple sources, evaluation indexed by the action that $\delta$ controls, and robustness of the signal–policy coupling under distribution and protocol shift.

debate systems expose peer confidence that can help or harm depending on how the protocol interprets it (Lin & Hooi, 2025). The open challenge is not just to fuse more signals, but to build provenance-aware confidence representations that keep track of where a belief comes from, detect when sources conflict, and determine which source should dominate which downstream decision.

**Challenge 4: Decision-aware objectives and evaluation.** One of the clearest lessons of this survey is that confidence quality cannot be summarized by a single metric. A signal that is useful for abstention is not necessarily optimal for conformal prediction; a signal that is well calibrated in isolation may still be poor for routing or adaptive compute allocation. Cost-aware routing systems optimize the cost–quality frontier rather than classical calibration (Chen et al., 2024c); conformal methods optimize coverage and set efficiency rather than ordinary pointwise confidence (Quach et al., 2024; Xi et al., 2025); abstention-oriented training shifts the refusal boundary itself rather than merely post-processing a fixed score (An & Xu, 2025); and PRM calibration matters precisely because an apparently good ranking score may still be a poor decision-grade success estimate (Park et al., 2025). A major open problem is therefore methodological: evaluation should be indexed by the downstream action that confidence governs. The field needs benchmarks and metrics that directly measure decision quality for filtering, stopping, routing, abstaining, covering, and aggregation, rather than assuming that one calibration metric can stand in for all of them. This argument now has a formal incentive-side statement and a first matching instrument: under binary grading, abstention is strictly suboptimal for any belief state, and nine of ten dominant benchmarks award zero credit to abstention (Kalai et al., 2025); conversely, integrating an answer-or-abstain utility over risk tolerances yields a proper decision-theoretic score that separates models which are indistinguishable under ECE and AURC, because rare high-confidence errors dominate abstention-aware risk (Wu et al., 2026). The converse case also exists: control roles that compare absolute probabilities across requests—such as the speculative-verification schedulers of §4—make calibration, not discrimination, the binding metric (Cheng et al., 2026). The survey's role axis suggests a concrete mapping. Selection should be scored by regret against an oracle reranker over the same candidate set; stopping by the samples or tokens needed to match fixed-budget accuracy (as in the budget-controlled comparisons of §4); routing and deferral by the area under the cost–quality deferral curve rather than by accuracy at one operating point; retrieval gating by retrieval calls saved at matched answer quality (§6); abstention by selective risk–coverage curves rather than ECE; coverage control by guaranteed-level validity together with set size or claim-retention efficiency (Quach et al., 2024; Xi et al., 2025); and aggregation by the calibration of the collective decision, not of individual voters (Chen et al., 2024b). A minimal decision-aware benchmark would therefore fix a task suite and a compute accounting, require each method to expose its signal $\kappa$, its transformation $T$, and its policy $\delta$ separately, and report the role-appropriate metric for each supported action—making it possible to say whether a signal that excels at abstention also transfers to routing on the same tasks.

**Challenge 5: Robustness and portability.** Confidence utilization methods are typically validated within a fixed model family, prompt format, task distribution, and control protocol. Whether the same signal remains meaningful after any of those ingredients change is much less understood. Verbal confidence is already known to be vulnerable under adversarial attack (Obadinma & Zhu, 2025), and black-box consistency

signals depend on how the model reacts to prompt perturbations (Joo et al., 2025). Formal guarantees inherit the same fragility: conformal-style thresholds certified under exchangeability visibly overshoot their risk targets under domain shift, on both the cascading and the abstention branches (Guo et al., 2026a; Dong & Shinnou, 2026). Model-selection systems also inherit a form of portability risk, since their learned or implicit confidence semantics depend on the available model pool and cost–quality frontier (Chen et al., 2024c). At the same time, work such as Superfiltering suggests that some useful control signals can transfer surprisingly well across scale (Li et al., 2024b). The open question is not whether transfer ever occurs, but when. A mature confidence-utilization framework should be able to detect when its assumptions no longer hold under domain shift, prompt shift, model replacement, adversarial manipulation, or protocol change, and then adapt or abstain accordingly.

Taken together, these challenges suggest that the next stage of the field is not simply to design more task-specific confidence heuristics, but to build confidence systems that are semantically interpretable, composable across units and horizons, source-aware, decision-aware, and robust to changing deployment conditions. Progress on these fronts would move confidence utilization beyond a collection of successful tricks and closer to a general theory of confidence as control in LLM systems.

## 10    Conclusion

This survey argued that confidence in LLM systems is most useful when treated not only as something to be estimated, but as something that changes behavior. Under the unified notation of Section 2, confidence is any reliability-relevant signal attached to a decision unit under a local decision state and then consumed by a downstream policy. Viewed through that lens, confidence utilization appears across six domains of the LLM lifecycle: training, inference, model selection and cascading, retrieval-augmented generation, risk management, and agentic control. In each domain, confidence matters because it governs actions such as filtering, selection, routing, retrieval, abstention, and search control.

The main lesson of the literature is not that there exists one best confidence signal, but that useful signals differ in semantics, source, unit, and objective. A score that works well for selection may be poor for abstention; a signal that is calibrated in isolation may still be misaligned with routing or retrieval; and a confidence estimate that is meaningful for an answer may not compose cleanly to trajectories, debates, or multi-stage systems. The field has therefore moved beyond confidence estimation alone. The harder problem is to build confidence systems that are interpretable, provenance-aware, composable across units and horizons, robust under shift, and evaluated by the quality of the decisions they support. Progress on these fronts would move the area beyond task-specific heuristics and toward a more general framework for reliable, efficient, and trustworthy control in LLM systems.

## 11    Limitations

This survey focuses on **confidence utilization** as a control signal rather than confidence estimation or calibration. Readers seeking comprehensive coverage of uncertainty quantification methods should consult complementary surveys.

We primarily survey English-language literature and methods evaluated on English benchmarks; confidence utilization in multilingual and low-resource settings remains comparatively underexplored.

Control policies inherit failure modes of the underlying confidence signal (miscalibration, bias, brittleness under distribution shift or adversarial prompting), which can lead to unsafe acceptance, premature deferral, or unnecessary escalation.

Many methods add latency or compute (sampling, verifiers, multi-stage retrieval), and formal guarantees (e.g., conformal prediction) depend on assumptions and proxies for correctness that may not hold under deployment drift.

Finally, our corpus inherits a publication-side selection bias: methods papers predominantly report settings in which confidence-gated control succeeds, so the evidence base likely understates how often such policies

fail or underperform simpler baselines. The negative and conditional results we surface in §4–§7 should be read with that asymmetry in mind.

**Implication.** Taken together, these limitations suggest that confidence-as-control should be viewed as a systems design paradigm rather than a drop-in solution: robust deployment requires validating confidence reliability under realistic shift, aligning proxy correctness with application risk, and accounting for compute and human factors alongside model accuracy.

## Broader Impact Statement

This survey studies how confidence signals can be used to control the behavior of large language model systems across training, inference, routing, retrieval, risk management, and agentic control. Better confidence utilization could yield meaningful societal benefits, including safer abstention, more reliable escalation to stronger models or human oversight, improved robustness in retrieval and reasoning pipelines, and more trustworthy deployment in high-stakes settings.

At the same time, confidence utilization introduces its own risks. Confidence signals are heterogeneous, imperfect, and often context-dependent; if they are treated as universally reliable, they may create a false sense of safety or justify incorrect automation decisions. In deployed systems, confidence-based filtering, refusal, routing, or prioritization may also encode hidden biases, suppress minority or out-of-distribution cases, or obscure responsibility when models defer to other components. We therefore view confidence not as a guarantee of correctness, but as a decision instrument whose meaning, robustness, and downstream consequences must be evaluated carefully in context.

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

# A   Related Surveys on Confidence, Calibration, and Hallucination

Several prior surveys examine confidence, uncertainty, calibration, hallucination, or abstention in LLMs. Their perspectives are complementary to ours, but their primary question is usually how a signal should be estimated, calibrated, diagnosed, or interpreted. By contrast, this survey asks how confidence-related signals are *used* to govern system behavior. Table 7 summarizes that distinction at a high level.

| Survey | Primary Focus | Confidence Perspective | Coverage | Relation to Ours |
|---|---|---|---|---|
| Geng et al. (2024) | Confidence estimation and calibration | Confidence as an object to estimate and calibrate | LLM confidence methods and evaluation | Estimator-centric; does not organize downstream control decisions |
| Xie et al. (2024) | Black-box calibration | Confidence elicitation, self-reflection, and recalibration without logits | Black-box LLM confidence calibration | Calibration-centric; narrower than lifecycle-wide control |
| Liu et al. (2025c) | Uncertainty quantification and calibration | Uncertainty decomposed into multiple sources | Multi-stage LLM UQ and evaluation | UQ-centric; complements our control view |
| Shorinwa et al. (2025) | Uncertainty quantification | Broad taxonomy of uncertainty sources and diagnostics | LLM uncertainty methods and tasks | UQ-centric; emphasizes estimation rather than action |
| Rawte et al. (2023) | Hallucination in foundation models | Detection and mitigation signals organized by failure type | Text, image, video, and audio foundation models | Failure-mode survey, not a confidence-utilization taxonomy |
| Huang et al. (2025b) | Hallucination in LLMs | Hallucination causes, detection, and mitigation | LLM lifecycle from pre-training to inference | Organized around one failure mode rather than one control signal |
| Tonmoy et al. (2024) | Hallucination mitigation | Intervention methods for reducing hallucination | LLM mitigation strategies, including RAG and decoding | Mitigation-centric; narrower than our control framing |
| Wen et al. (2025) | Abstention and "knowing what you don't know" | Confidence as a basis for refusal or deference | Pre-training, fine-tuning, alignment, and inference | Centered on one decision family |
| Zhang et al. (2026) | Evolution of UQ from metric to signal | Uncertainty as an active control signal | Reasoning, autonomous agents, and reinforcement learning | Closest to ours in thesis; frontier-organized rather than lifecycle-wide (see below) |
| **This survey** | **Confidence utilization** | **Confidence as a control signal** | **Training, inference, routing, RAG, risk, and agentic control** | **Cross-domain framework for confidence-guided decisions** |

Table 7: Related surveys on confidence, uncertainty, hallucination, and abstention in LLMs. Prior surveys mainly organize the literature around estimation, calibration, failure analysis, or a single decision family. This survey instead treats confidence-related signals as control inputs that govern downstream system behavior across the LLM lifecycle.

The most direct predecessors are the confidence-estimation and uncertainty-quantification surveys. Geng et al. (2024) review confidence estimation and calibration for LLMs, while Xie et al. (2024) focus specifically on black-box calibration settings where internal logits may be unavailable. Liu et al. (2025c) and Shorinwa et al. (2025) broaden the discussion to uncertainty quantification, distinguishing different uncertainty sources and surveying calibration diagnostics, conformal tools, and related evaluation methods. These surveys provide essential background on how confidence-like signals are elicited, estimated, or calibrated, but they do not systematically trace how such signals are consumed by downstream policies once they are available. In the language of Section 2, they focus primarily on signal formation, whereas our emphasis is on the actions those signals support.

A second line of related work organizes the literature around hallucination rather than confidence. Rawte et al. (2023) survey hallucination across foundation models and modalities; Huang et al. (2025b) provide an LLM-specific view of hallucination causes, benchmarks, detection, and mitigation across the lifecycle; and Tonmoy et al. (2024) emphasize mitigation strategies such as retrieval, decoding constraints, and fine-tuning interventions. These surveys are highly relevant to our discussions of groundedness, failure detection, and abstention, especially in Section 6 and Section 7. However, they are organized around a particular failure mode. Our survey instead treats hallucination detection as one instance of a broader pattern in which confidence-related signals trigger decisions such as filtering, retrieval, abstention, escalation, or revision.

Closest to this survey in thesis is the concurrent work of Zhang et al. (2026), who chart the "functional evolution" of uncertainty quantification from a passive, post-hoc metric into an active control signal. The shared motivation is genuine, and we point readers to their treatment of uncertainty-robust reward modeling and multi-agent uncertainty propagation, which we cover only briefly. The two surveys are nonetheless organized around different objects. First, they organize by application frontier—reasoning, autonomous agents, and reinforcement learning—whereas we organize by lifecycle stage under an operational source/unit/role taxonomy with explicit inclusion criteria (C1–C3) and access-level annotations, so a practitioner can determine whether a method is deployable under given API constraints. Second, two of our six domains—model routing and cascading, and retrieval-augmented generation—are absent from their taxonomy, and risk management appears there through abstention thresholds and conformal theory rather than as a standalone operational domain organized around deployment decisions and guarantee units; they also explicitly place estimation and calibration out of scope, whereas we treat calibration-for-control as a first-class concern, since some control roles consume absolute probabilities and therefore inherit calibration requirements (§4, §9). Third, they identify the misalignment between static evaluation protocols and dynamic control needs as an open challenge; our Challenge 4 operationalizes that observation into a decision-aware evaluation protocol with role-matched metrics.

Wen et al. (2025) are closest in spirit to our risk-management discussion because they treat abstention as a first-class action and analyze how models can decide when not to answer. Their contribution is important precisely because it shifts attention from confidence estimation alone to a downstream decision. Our framing extends that idea in two ways. First, we treat abstention as one member of a broader family of confidence-guided actions that also includes selection, routing, retrieval control, search control, and aggregation. Second, we follow those actions across six parallel domains of the LLM lifecycle rather than centering a single decision family.

**Summary.** Prior surveys provide strong foundations on confidence estimation, uncertainty quantification, hallucination analysis, and abstention. Our contribution is to connect these strands through a unified control perspective: confidence-related signals are not only quantities to be estimated, but instruments for governing behavior across the full LLM lifecycle.

