# OpenReview forum: "Confidence as Control: A Survey of Confidence Utilization in Large Language Models"
_TMLR — Under review for TMLR_

### Review · Reviewer_Cmcz · 2026-06-10

**Summary Of Contributions:**

The survey studies confidence utilization at different levels of the LLM lifecycle including training, inference, model selection and cascading,
retrieval-augmented generation, risk management, and agentic control. The survey focuses on how confidence is used as a control primitive at each of these stages, rather than on how the confidence score is estimated.

**Strengths**

1. The paper makes a convincing case that confidence in LLMs should not only be studied as an estimation or calibration problem, but as a control signal that governs system behavior.

2. The decision-state, decision-unit, confidence-signal, transformation, and policy formulation gives the survey a coherent structure and helps compare otherwise very different methods.

3. The paper covers confidence utilization across training, inference, model routing, RAG, risk management, and agentic systems, giving readers a useful map of a rapidly growing area.

**Weaknesses**
1. The paper adopts a very broad operational definition of confidence. While this helps unify diverse methods, it also blurs the boundary between confidence, uncertainty, utility, difficulty, quality, and reward. As a result, some included methods appear only confidence-adjacent, and the central concept risks becoming less precise.

2. The paper would benefit from a clearer explanation of search strategy, inclusion and exclusion criteria, and how borderline or confidence-adjacent papers were selected.

3. Some sections read more like a catalog of methods than a critical synthesis. The paper could more clearly compare which signals work best for which control decisions, under which assumptions, and with what limitations.

4. It does not discuss relevant foundational literature on confidence utilization in machine learning (see requested changes below).

**Additional Comments:**

None.

**Audience:**

Yes

**Audience Explanation:**

Yes. The paper is likely to interest at least part of the TMLR audience, especially researchers working on LLM reliability, uncertainty estimation, calibration, selective prediction, retrieval-augmented generation, model routing, and agentic systems. Its main value is not a new empirical finding, but a useful conceptual synthesis: it reframes confidence as a control signal that governs downstream decisions across the LLM lifecycle. This perspective should be relevant to readers interested in building more reliable and adaptive LLM systems.

**Broader Impact Concerns:**

None.

**Claims And Evidence:**

Yes

**Claims Explanation:**

The main claims are mostly supported by relevant and clearly presented evidence, especially through its broad synthesis of confidence utilization across training, inference, routing, RAG, risk management, and agentic control. However, the support is stronger as a conceptual taxonomy than as a systematic survey, since the paper does not clearly specify its literature search strategy, inclusion criteria, or treatment of borderline cases. The evidence would be more convincing if the authors more sharply justified why confidence-adjacent notions such as utility, difficulty, reward, and routing suitability belong under the same “confidence-as-control” framework.

**Requested Changes:**

1. **Clarify the survey methodology. Critical.**
   Specify the search strategy, inclusion and exclusion criteria, time window, and how borderline papers were handled. This is important for assessing coverage and reproducibility.

2. **Sharpen the definition of confidence. Critical.**
   The current definition is very broad and sometimes blurs confidence with utility, difficulty, quality, reward, or routing suitability. The paper should more clearly justify what counts as confidence utilization.

3. **Distinguish core from adjacent work. Critical.**
   Some included methods seem only indirectly related to confidence. Marking methods as core, adjacent, or background would make the taxonomy easier to interpret.

4. **Add more comparative synthesis. Strengthening.**
   Several sections read like method catalogs. The paper would benefit from clearer comparisons of which confidence signals work best for which control decisions and under which assumptions.

5. **Develop the evaluation discussion. Strengthening.**
   The authors should make the decision-aware evaluation argument more actionable, for example by mapping control roles such as routing, abstention, retrieval, stopping, and aggregation to suitable metrics.

6. **Strengthen links to foundational literature where confidence is often used. Strengthening.**
   The paper would benefit from stronger connections to selective prediction, decision theory, optimal stopping, active learning, active feature acquisition, cascades, bandits, POMDPs, and risk-sensitive control.

---

> ### Author Response · Authors · 2026-07-03
> **Response to Reviewer Cmcz**
>
> We thank the reviewer for the balanced assessment and for the concrete, prioritized list of requested changes. All three Critical items and all three Strengthening items have been implemented.
>
> ---
>
> > Critical 1: "Clarify the survey methodology"
>
> §2.2 now contains a dedicated methodology paragraph specifying: (i) **corpus construction** in three stages — seeding from the estimation/calibration surveys (Appendix A) by extracting all cited works with a downstream-decision contribution; keyword search over the ACL Anthology, OpenReview (ICLR/ICML/NeurIPS/TMLR), and arXiv, crossing signal vocabulary (confidence, uncertainty, calibration, semantic entropy, verifier, process reward) with action vocabulary (selection, filtering, routing, cascading, deferral, abstention, adaptive retrieval, early stopping, backtracking, debate); and forward/backward snowballing of all included papers; (ii) **inclusion criteria** — every candidate scored against an explicit three-part test C1–C3 (reliability semantics, decision coupling, state locality); (iii) **borderline handling** — retained with an explicit "adjacent" marker and discussed in the text where they appear; (iv) **time window** — emphasis on 2022 onward, when LLM-specific confidence signals became distinct from classical calibration; and (v) an honest statement that the corpus is criteria-driven and curated rather than exhaustive, prioritizing methods that introduce a new signal–policy coupling.
>
> >Critical 2: "Sharpen the definition of confidence"
>
> Two changes. First, the formal definition now attaches an explicit **target property** ρ(u) to every signal: κ is required to be interpretable as a monotone estimate of Pr[ρ(u)=1 | ξ], and stating ρ explicitly is what separates confidence from utility, difficulty, quality, reward, and routing suitability — these are distinct target properties, not interchangeable notions. Second, §2.2 disambiguates the four senses in which the literature uses the word (estimated correctness / expected utility / verifier-derived quality / control value) and locates each in the framework. The framework's breadth is now explicitly framed as unifying the *control interface* rather than asserting shared semantics.
>
> >Critical 3: "Distinguish core from adjacent work"
>
> Done, comprehensively: every row of all six comparison tables is now marked ● (core: C1–C3 satisfied with a reliability-reading target property) or ○ (adjacent: same control problem, but the target property is utility, difficulty, influence, quality, or reward), with the convention explained in each caption and the criteria in §2.2. The markers make the taxonomy's boundary judgments auditable at a glance.

---

> ### Author Response · Authors · 2026-07-03
>
> >Strengthening 1: "Add more comparative synthesis"
>
> For the revision we read the full PDFs of 35 core papers in the inference, RAG, and risk sections and rewrote those sections' syntheses around exact numbers, empirical orderings, and head-to-head conflicts. Representative additions: the empirical ordering for inference-time candidate selection (PRM > confidence-weighted voting > majority voting > sequence probability, with supporting numbers); evidence that within-question discrimination rather than calibration predicts selection gains; the finding that internal-state gating dominates multi-hop retrieval decisions while trained pre-retrieval gates win single-hop efficiency (with EM and retrieval-call numbers); a documented non-transfer (contrastive decoding hurting GSM8K in DoLa's controlled comparison, while the original CD paper contains no reasoning experiments); detector rankings shown to be benchmark- and estimator-conditional (AUROC comparisons); and the demonstration that ECE-improving calibration can enlarge conformal sets. Each subsection in these sections now closes with a when-does-this-work / failure-mode note drawn from the papers' stated limitations.
>
> >Strengthening 2: "Develop the evaluation discussion... map control roles to suitable metrics"
>
> Challenge 4 now contains exactly this mapping: selection → regret against an oracle reranker over the same candidates; stopping → samples/tokens needed to match fixed-budget accuracy; routing and deferral → area under the cost–quality deferral curve; retrieval gating → retrieval calls saved at matched answer quality; abstention → selective risk–coverage curves; coverage control → validity at the guaranteed level together with set-size/claim-retention efficiency; aggregation → calibration of the collective decision rather than of individual voters. It further sketches a minimal decision-aware benchmark protocol (fixed task suite and compute accounting; each method exposes κ, T, δ separately; role-appropriate metrics reported per supported action).
>
> >Strengthening 3: "Strengthen links to foundational literature"
>
> A new "Historical roots" paragraph in §2 connects the framework to the pre-LLM literature the reviewer listed, with citations added: the rejection option (Chow, 1970) and selective prediction (El-Yaniv & Wiener, 2010; Geifman & El-Yaniv, 2017) as ancestors of the abstention role; learning-to-defer (Madras et al., 2018; Mozannar & Sontag, 2020) anticipating cascade deferral; classifier cascades (Viola & Jones, 2001) as direct ancestors of model cascading; sequential analysis and optimal stopping (Wald, 1945) underlying adaptive-sampling rules; active learning (Settles, 2009) prefiguring confidence-aware data selection; and bandit/POMDP formulations (Lattimore & Szepesvári, 2020; Kaelbling et al., 1998) supplying the decision-theoretic language for routing under partial observability — noting that AutoMix's POMDP-based deferral already instantiates this connection. The paragraph also observes that decision-aware evaluation (Challenge 4) has well-developed analogues in this older literature that the LLM community has only begun to import.
>
> We thank the reviewer again; the requested changes materially improved the paper's rigor and usability.

---

### Review · Reviewer_9R5k · 2026-06-10

**Summary Of Contributions:**

The paper surveys the LLM literature on confidence utilization: how reliability-related signals are used to govern system behavior, as distinct from the larger body of work on confidence estimation and calibration. It proposes a unified notation in which a confidence signal is defined over decision units u under a local decision state, optionally transformed, and consumed by a policy to produce an action, together with a three-axis decomposition by signal source, decision unit/granularity, and functional role. Using this framework, it organizes methods across six lifecycle domains (training, inference, model selection/cascading, RAG, risk management, agentic control), with per-domain comparison tables and a consolidated taxonomy, and concludes with five cross-cutting open challenges.

**Audience:**

No

**Audience Explanation:**

I note that the paper's interest is primarily as an organized reference rather than as a source of new insight.

**Claims And Evidence:**

No

**Claims Explanation:**

- The unifying framework is too permissive to unify anything. The survey's core claim is that a single abstraction provides "a common language that makes the design space explicit." But the framework is defined by deliberately maximal breadth confidence is "a probability-like estimate, an uncertainty score, a log-probability-derived quantity, a verbalized confidence value, a sample-agreement statistic, a semantic uncertainty measure, a verifier or judge score, a reward-model score, or a hybrid", and a unit may be "a token, span, claim, retrieved chunk, training example, candidate response, model, tool, reasoning step, trajectory, or agent vote." Under definitions this elastic, essentially every method in the literature fits, so the abstraction describes rather than organizes. The paper itself concedes this in Challenge 1 (heterogeneous semantics): the signals "are not obviously commensurate." That admission undercuts the framing: if the signals are not commensurate, then mapping them onto a shared κ does not actually unify them; it relabels heterogeneous quantities with a common symbol while their meanings remain incompatible. The survey never resolves the tension between its claimed unification and its own acknowledgment that the unified object has no shared semantics.

- The framework is descriptive, not providing any insights. A useful organizing framework should yield at least one non-obvious consequence: an empty cell in the source × unit × role space, a predicted method, a reclassification that changes how a reader understands a family of work, or a contradiction made visible only through the shared notation. The survey produces none of these. The notation is introduced in §2 and thereafter mostly restated per section ("in the notation of §2, ξ now contains…") without generating insight that the section's prose would not have conveyed unaided. The axes function as table column headers rather than as an analytical instrument.

- Synthesis is taxonomically sharp but rather thin from the evaluation perspective. For a survey whose contribution is comparative organization, there is little independent assessment of which methods work, under what conditions, or where reported results conflict. Method characterizations largely restate each paper's own framing. The reader rarely learns which approaches are empirically dominant, which fail to reproduce, or how results compare across the works grouped in the same table cell: precisely the synthesis a reader cannot get from the primary sources themselves. The multi-agent subsection (§8.4), which does surface contradictory findings, shows what the rest of the survey could have been and highlights the gap elsewhere.

- The trustworthiness payoff is asserted, not demonstrated. The abstract and introduction promise a path to "more reliable and trustworthy LLM systems," but the survey is explicitly descriptive and never connects its taxonomy to concrete reliability outcomes. The framing oversells what an organizational survey delivers.

- Conceptual figures carry little information. Figures 1 and 4 are stylized illustrations that restate prose without adding content (clearly AI-generated with little insights); Figure 4 in particular re-renders the five challenges pictorially. The substantive visual contribution is confined to the tables and the taxonomy (Fig. 3). These figures should be made information-bearing: e.g., a figure that actually populates the source/unit/role space with method clusters and exposes its gaps (or should be removed).

**Requested Changes:**

Critical (needed to support the central claim):

See claims section. Also as follows:

- Demonstrate that the framework does analytical work. Concretely: populate the source × unit × role space and identify at least one under-explored or empty region, OR show a case where the shared notation makes a cross-domain contradiction or equivalence visible that the per-paper framing obscures. Without this, "unified framework" should be downgraded to "organizing taxonomy" throughout.

- Add evaluative synthesis to at least the inference, RAG, and risk sections: where results conflict, where methods fail to transfer, and which signals are empirically preferred for which action. State the paper selection criteria (how the corpus was chosen; systematic vs. curated).

Recommended:

- Soften the "trustworthy/reliable systems" framing or tie it to specific reliability results.
- Rework or remove Figures 1 and 4.

---

> ### Author Response · Authors · 2026-07-03
> **Response to Reviewer 9R5k**
>
> We thank the reviewer for an thorough and candid review. It identified the single most important weakness of the submission: the framework was *defined* but not *used*. The revision addresses this directly and substantially. Below we respond point by point; we accept most of the criticism and act on it, and we respectfully push back on one specific inference.
>
> >1. "The unifying framework is too permissive to unify anything" / "relabels heterogeneous quantities with a common symbol"
>
> We accept part of this criticism and contest part of it.
>
> **What we accept.** In the submitted version, the notation was introduced in §2 and then mostly restated per section without generating conclusions — the reviewer's description "axes function as table column headers" was fair. The revision fixes this in the way the reviewer requested (see Point 2).
>
> **What we contest.** The inference "the signals are not commensurate, therefore mapping them onto a shared κ does not unify them" conflates two different kinds of unification. Our framework unifies the *control interface* — how a reliability-relevant score enters a decision (signal → transformation → policy → action) — and explicitly does not claim the signals share probabilistic semantics. This is now stated in §2 (first paragraph): the abstraction unifies problems in the same way a Markov decision process unifies problems whose rewards measure incommensurable quantities; no one considers MDPs "relabeling" because reward semantics differ across applications. To make the semantic heterogeneity precise rather than elastic, the revised definition attaches an explicit *target property* ρ(u) to every signal (§2.1), and §2.2 disambiguates the four senses in which the literature uses "confidence." Under this reading, Challenge 1 is not a concession that undercuts the framework; it is a *finding the framework produces*: because policies consume κ̃ through one interface, semantically different signals become interchangeable at the point of control even though they should not be — a failure mode that is invisible without the shared interface. We have kept the challenge and sharpened its statement accordingly.
>
> >2. "Demonstrate that the framework does analytical work: populate the source × unit × role space and identify an empty region, OR show a cross-domain contradiction or equivalence"
>
> We did both.
>
> - **Populated design space (new Figure 3).** We aggregated all six comparison tables into a source × role matrix (with units annotated in the per-domain tables). The matrix identifies unexplored regions we believe are genuine design opportunities: external-evidence signals are never consumed as learning signals; peer-communicated confidence is unused outside multi-agent deliberation; token-granularity control in model selection exists only as speculative hand-off. It also exposes a *structural* constraint: sample-agreement and answer-conditioned sources are unavailable to pre-call decisions for timing reasons — which explains, rather than merely lists, why pre-call routers converge on auxiliary predictors while post-hoc deferral converges on self-signals.
> - **Cross-domain equivalences (§2.3).** Cascade deferral (§5), risk-management abstention (§7), and retrieval triggering (§6) instantiate the same policy template a = GET-HELP if κ̃ < τ, differing only in what supplies the help; consequently, threshold-selection and calibration techniques transfer across these literatures, and the answer-conditioned vs. answer-free timing split from routing reappears verbatim in the when-to-retrieve literature under different names. Adaptive-consistency stopping and cascade stopping are likewise the same optimal-stopping problem over different unit sequences.
> - **Contradictions exposed (§2.3).** Three, each developed quantitatively in the body: (i) the same comparison κ̃ < τ triggers *opposite* actions at training vs. deployment time; (ii) calibration maps that improve pointwise metrics can strictly worsen conformal set efficiency (ECE improves while APS set size grows 4.91→6.69; §7); (iii) verbalized confidence is simultaneously among the better-calibrated signals for RLHF answering and among the worse ones in evaluator pipelines, which the framework attributes to a change in target property ρ rather than an inconsistency in the literature.
>
> Given that the revision now does the analytical work the reviewer asked for, we have retained the term "unified framework," and we hope the reviewer will agree the term is now earned rather than asserted.

---

> ### Author Response · Authors · 2026-07-03
>
> >3. "Synthesis is taxonomically sharp but thin from the evaluation perspective"
>
> We agree this was the second major gap. For the revision we read the full PDFs of 35 core papers in the inference, RAG, and risk sections and added evaluative synthesis with exact numbers throughout those sections, in the style the reviewer praised in §8.4. Examples now in the text:
>
> - **Inference (§4):** the empirical ordering for candidate selection (PRM 78.2% > ORM 72.4% > majority 69.6% on MATH at large budgets; confidence-weighted voting recovering majority-vote accuracy at 46% lower cost); the finding that *within-question discrimination*, not calibration, predicts selection gains (the best-calibrated signal, ECE 0.005, is the worst selector); and a documented non-transfer: the claim that contrastive decoding helps reasoning does not originate in the cited CD paper (which contains no reasoning experiments), and DoLa's controlled comparison shows CD *hurting* GSM8K (33.8→28.4 at 33B).
> - **RAG (§6):** internal-state gating (SEAKR) dominates multi-hop under an identical pipeline (2Wiki EM 30.2 vs 22.4/14.3/4.6 for DRAGIN/FLARE/Self-RAG) while trained pre-retrieval gates win single-hop efficiency (Adaptive-RAG matches multi-step EM at 1.03 vs 2.81 steps); reflection tokens collapse on multi-hop (Self-RAG MuSiQue EM 1.60); and the axiomatic analysis of Soudani et al. is developed into the section's central negative result, with an explicit list of which method families it undercuts and which it spares.
> - **Risk (§7):** head-to-head AUROC comparisons across detector families, including the estimator- and benchmark-conditional ranking flip (semantic entropy 0.790 under one protocol vs. 63–65 under another while EigenScore reaches 71–73); precise statements of what unit each conformal method guarantees and the practical consequence; and the train-time-vs-post-hoc abstention comparison including a metric-driven sign flip (R-Tuning wins under AP, loses under AED).
>
> Each subsection in these sections now ends with a when-does-this-work / failure-mode note grounded in the papers' own stated limitations.
>
> >4. "State the paper selection criteria"
>
> §2.2 now specifies the corpus construction (survey seeding, venue/keyword search families, forward/backward snowballing), the explicit inclusion test C1–C3 applied to every candidate, the handling of borderline cases (retained with an "adjacent" marker, now applied in all six comparison tables), and the curation policy. We describe the corpus honestly as criteria-driven and curated rather than exhaustive.
>
> >5. "The trustworthiness payoff is asserted, not demonstrated"
>
> We agree the framing overreached. The abstract now ties the claim to the concrete, measured outcomes the survey documents (cost reduction at matched accuracy, coverage guarantees, answer-rate/error-rate trade-offs) and positions principled confidence use as a prerequisite for reliability rather than a guarantee of it. The Broader Impact statement retains the cautionary discussion.
>
> >6. "Conceptual figures carry little information"
>
> Both figures have been replaced with information-bearing vector graphics. Figure 1 is now a TikZ diagram that maps each lifecycle stage to the specific control actions confidence drives there (with section pointers); the former Figure 4 is a compact strip stating each challenge in the framework's notation; and the new Figure 3 populates the design space as the reviewer requested. No raster illustrations remain.
>
> ---
> ## On "Would TMLR's audience be interested" (No)
>
> We hope the revision changes this assessment: beyond the organized reference the reviewer acknowledged, the paper now offers empirical orderings and documented non-transfers that practitioners cannot get from primary sources individually, a populated design space with identified gaps, and cross-domain transfers of technique (e.g., cascade calibration methods applying to retrieval gating) that require exactly the shared abstraction the paper contributes.
>
> We again thank the reviewer — the revision is substantially stronger because of this review.

---

### Review · Reviewer_qciC · 2026-06-15

**Summary Of Contributions:**

This paper surveys confidence utilization in large language models, shifting the focus from confidence estimation, uncertainty quantification, and calibration to how confidence signals are used to control system behavior. The authors propose a unified confidence-as-control framework in which confidence is defined over decision units under a local decision state and is consumed by a policy to determine downstream actions. Using this lens, the paper organizes prior work across the LLM lifecycle, including training, inference, model selection and cascading, retrieval-augmented generation, risk management, and agentic control. The paper further compares methods by signal source, decision unit, and functional role, and identifies open challenges in confidence semantics, composition, source attribution, decision-aware evaluation, and robustness.

Strengths:
- The paper addresses an important and timely problem: how confidence signals should be used in deployed LLM systems rather than only how they should be estimated or calibrated.

- The proposed confidence-as-control perspective is useful and helps connect several otherwise separate areas, including selective generation, adaptive inference, model routing, RAG, conformal control, and agentic decision making.

- The unified notation based on decision state, decision units, confidence signals, transformations, policies, and actions provides a clear abstraction for comparing methods that operate at different granularities, such as tokens, claims, retrieved chunks, candidate answers, models, trajectories, and agent votes.
- From my perspective, The discussion of open challenges is meaningful. The paper correctly identifies that confidence quality should be evaluated with respect to the downstream decision it controls, rather than through a single generic calibration metric.

Weaknesses:

- The definition of confidence is very broad, covering uncertainty scores, log-probability-based signals, verbalized confidence, sample agreement, semantic entropy, verifier scores, judge scores, reward-model scores, and expected utility signals. While this broad scope helps unify many papers, it also makes the boundary between confidence utilization and general score-based system control somewhat unclear.

- The paper sometimes includes methods that are only confidence-adjacent, such as data quality scoring, influence-based data selection, utility prediction, or reward-model scoring. The authors do acknowledge this issue in several places, but the criteria for inclusion could be made more explicit and consistently applied.

- The paper would benefit from a clearer distinction between confidence as estimated correctness, confidence as expected utility, confidence as verifier-derived quality, and confidence as control value. Without this distinction, some readers may find the central concept overloaded.

**Audience:**

Yes

**Audience Explanation:**

The paper is relevant to researchers working on LLM reliability, uncertainty estimation, calibration, RAG, model routing, selective generation, test-time computation, and agentic AI. It would also be useful for researchers building practical LLM systems where confidence signals are used to decide when to retrieve, defer, abstain, rerank, verify, retry, or escalate to stronger models. The survey is especially useful for readers who already know confidence estimation literature but want a broader view of how such signals are used in full LLM systems.

**Claims And Evidence:**

No

**Claims Explanation:**

The main claims are supported by a broad and well-organized review of existing literature. The paper clearly demonstrates that confidence-related signals are already being used to control training, inference, routing, RAG, risk management, and agentic behavior. The unified framework is also supported by many examples across these domains.

However, because the paper is a survey rather than an empirical study, some higher-level claims are supported more by conceptual synthesis than by direct evidence. In particular, the claim that confidence should be treated as a general control primitive is convincing as a framing, but the paper does not provide empirical evidence that this framing improves benchmark design, system performance, or method choice

**Requested Changes:**

- The paper should more clearly define the boundary between confidence utilization and general score-based control. For example, the authors should explain when reward scores, judge scores, quality scores, utility estimates, or influence scores should count as confidence signals and when they should be considered adjacent but outside the main scope.

- The discussion of agentic control could be expanded. Since confidence-guided tool use, backtracking, verifier-based search, and multi-agent voting are increasingly important, the paper would benefit from a deeper treatment of how confidence signals compose across multiple agent steps.

- The open challenges section is strong, but it could be made more actionable by proposing concrete benchmark designs or evaluation protocols for decision-aware confidence evaluation.

- The authors should strengthen the positioning relative to existing surveys on uncertainty quantification, calibration, selective prediction, adaptive computation, model routing, and RAG. The paper currently states that prior surveys focus mainly on estimation and calibration, but the distinction from related work could be made more systematic.

- typo: “Fig. 4 ,” has an extra space before the comma. It should be “Fig. 4,”.
- consistency: The paper alternates between “confidence-as-control,” “confidence-guided control,” and “confidence utilization.” These are related but not identical. The authors should clarify whether these terms are interchangeable or whether they refer to different levels of abstraction.

---

> ### Author Response · Authors · 2026-07-03
> **Response to Reviewer qciC**
>
> We thank the reviewer for the careful reading and the constructive assessment. All requested changes have been implemented; we detail each below.
>
> ---
>
> >1. "More clearly define the boundary between confidence utilization and general score-based control"
>
> The revised §2.2 ("Scope: What Counts as Confidence Utilization") introduces an explicit three-part inclusion test applied to every method in the survey:
>
> **C1 (reliability semantics):** the signal must be interpretable as a monotone estimate of a stated target property ρ(u) concerning the reliability, correctness, groundedness, or expected success of the unit;
>
>   - **C2 (decision coupling):** the signal must be consumed by a policy that changes system behavior;
>   - **C3 (state locality):** the signal must be computed online for the current unit and state.
>
> This directly answers the reviewer's question about when reward, judge, quality, utility, or influence scores count: methods whose target property is expected utility, difficulty, data influence, generic quality, or reward satisfy C2–C3 but not C1's reliability reading; they are retained as **adjacent** and now explicitly marked ○ (vs. ● for core) in *all six* comparison tables, with the criteria referenced in each caption. Estimation-only work fails C2 (background); offline corpus statistics fail C3 (excluded).
>
> Relatedly, the revision distinguishes the four senses the reviewer asked about — confidence as estimated correctness, as expected utility, as verifier-derived quality, and as control value — and specifies how they map onto the framework: the first three are semantic families of the raw signal κ, while "control value" is the post-transformation quantity κ̃ that exists in every method (§2.2, last paragraph).
>
> > 2. "The discussion of agentic control could be expanded... how confidence signals compose across multiple agent steps"
>
> The Discussion of §8 now contains a substantially deeper treatment of composition. It formalizes the typical agent as a stack of three signal–policy couplings (escalation trigger κ⁽¹⁾, step/search signal κ⁽²⁾, aggregation signal κ⁽³⁾) and derives the key difficulty: each downstream signal is evaluated only on units that survived the upstream gate, so its effective input distribution is a selection-biased residual — which compounds the out-of-distribution overconfidence already documented for process reward models. The same argument explains, in the reverse direction, why exposing per-agent confidence does not reliably improve debate outcomes: upstream self-correction loops modify each agent's confidence distribution differently, so communicated confidences are not a common currency. We close with three concrete directions (joint calibration on post-selection distributions; propagation of the controller's own second-order uncertainty; protocol-aware aggregation).
>
> >3. "The open challenges section... could be made more actionable by proposing concrete benchmark designs or evaluation protocols"
>
> Challenge 4 now includes (i) an explicit role→metric mapping derived from the survey's role axis — selection scored by regret against an oracle reranker; stopping by samples/tokens to match fixed-budget accuracy; routing by area under the cost–quality deferral curve; retrieval gating by retrieval calls saved at matched answer quality; abstention by selective risk–coverage curves; coverage control by validity plus set-size efficiency; aggregation by calibration of the collective decision — and (ii) a minimal decision-aware benchmark protocol: fix a task suite and compute accounting, require each method to expose κ, T, and δ separately, and report the role-appropriate metric for each supported action, making cross-role transfer of a signal measurable for the first time.
>
> >4. "Strengthen the positioning relative to existing surveys... more systematic"
>
> Three additions: (a) the appendix comparison (Table + prose) now distinguishes each prior survey along the survey's own axes rather than only by topic; (b) a new "Historical roots" paragraph in §2 systematically connects the framework to the pre-LLM literature where confidence already acted as control — rejection option (Chow, 1970), selective prediction, learning-to-defer, classifier cascades, sequential analysis/optimal stopping, active learning, bandits and POMDPs — stating for each which unit/role in our decomposition it anticipates; (c) the introduction now states explicitly what distinguishes this survey from a catalog (stated inclusion criteria; analytical use of the decomposition; the populated design space of the new Figure 3).

---

> ### Author Response · Authors · 2026-07-03
>
> >5. Typo and terminology consistency
>
> - The "Fig. 4 ," spacing typo is fixed.
> - §2 now opens with a Terminology paragraph fixing the three terms at three levels of abstraction: *confidence utilization* (the object of study), *confidence-as-control* (the framework), and *confidence-guided X* (a specific mechanism instantiating the pipeline in one domain). All standalone uses of "confidence-guided control" in running prose have been normalized accordingly; adjectival uses in section titles refer to specific mechanisms, consistent with the definition.
> ---
> ## On the evidence question ("conceptual synthesis rather than direct evidence")
>
> While the paper remains a survey, the revision substantially strengthens its evidential character: we read 35 core papers in full and added quantitative, evaluative synthesis (empirical orderings, head-to-head conflicts, and failure-mode notes with exact numbers) throughout §4, §6, and §7, and the claim that the framing itself is productive is now supported concretely — the new Figure 3 identifies unexplored regions of the design space, and §2.3 derives cross-domain equivalences (e.g., cascade deferral ≡ abstention ≡ retrieval triggering) with practical transfer consequences.
>
> We thank the reviewer again for feedback that materially improved the paper.